# In vivo genome editing via CRISPR/Cas9-mediated homology-independent targeted integration for Bietti crystalline corneoretinal dystrophy treatment

Xiang Meng [1,2], Ruixuan Jia[1,2], Xinping Zhao[3], Fan Zhang[3], Shaohong Chen[3], Shicheng Yu[1,2], Xiaozhen Liu[1,2], Hongliang Dou[1,2], Xuefeng Feng[1,2], Jinlu Zhang[3], Ni Wang[3], Boling Xu[1,2] & Liping Yang [1,2] ✉

Bietti crystalline corneoretinal dystrophy (BCD) is an autosomal recessive chorioretinal degenerative disease without approved therapeutic drugs. It is caused by mutations in *CYP4V2* gene, and about 80% of BCD patients carry mutations in exon 7 to 11. Here, we apply CRISPR/Cas9 mediated homology-independent targeted integration (HITI)-based gene editing therapy in HEK293T cells, BCD patient derived iPSCs, and humanized *Cyp4v3* mouse model (*h-Cyp4v3^{mut/mut}*) using two rAAV2/8 vectors via sub-retinal administration. We find that sgRNA-guided Cas9 generates double-strand cleavage on intron 6 of the *CYP4V2* gene, and the HITI donor inserts the carried sequence, part of intron 6, exon 7-11, and a stop codon into the DNA break, achieving precise integration, effective transcription and translation both in vitro and in vivo. HITI-based editing restores the viability of iPSC-RPE cells from BCD patient, improves the morphology, number and metabolism of RPE and photoreceptors in *h-Cyp4v3^{mut/mut}* mice. These results suggest that HITI-based editing could be a promising therapeutic strategy for those BCD patients carrying mutations in exon 7 to 11, and one injection will achieve lifelong effectiveness.

Bietti crystalline corneoretinal dystrophy (BCD; OMIM 210370) is an autosomal recessive progressive chorioretinal degenerative disease, characterized by small yellow-white glittery crystalline deposits throughout the posterior pole of the retina[1]. Clinically, most BCD patients suffer progressive nyctalopia, constriction of the visual field, and even decreased visual acuity since the second and third decade of life, and progress to legal blindness by the fifth or sixth decade of life[2,3]. The prevalence of BCD was estimated to be 1/67,000 in Europe[4]. However, it is more common in East Asia, estimated to be 1/25,000 in China, and is one of the most common pathogenic genes in Chinese inherited retinal dystrophies (IRDs) patients[5,6].

*CYP4V2* (Gene ID 285440, OMIM 210370) is the only identified gene associated with BCD, it comprises 11 exons and encodes a protein of 525 amino acids. The CYP4V2 protein is an ω-3-polyunsaturated hydroxylase, and the endogenous substrates of CYP4V2 are ω-3-polyunsaturated fatty acids, such as docosahexaenoic acid (DHA) and eicosapentaenoic acid (EPA)[7,8]. The photoreceptor outer segment contains large amounts of lipids that can be esterified and transformed into DHA, EPA, and other types of fatty acids. In a physiological state, retinal pigment epithelium (RPE) engulfs the shedding outer segment, the lipids in the disks are metabolized in RPE cells and transferred to the inner segment for biosynthesis of the new disks. Lipid recycling

[1]Department of Ophthalmology, Third Hospital, Peking University, Beijing, China. [2]Beijing Key Laboratory of Restoration of Damaged Ocular Nerve, Peking University Third Hospital, Beijing, China. [3]Beijing Chinagene Co., LTD, Beijing, China. ✉e-mail: alexlipingyang@bjmu.edu.cn

promotes disks regeneration and maintains the function of photoreceptors. Mutations in the *CYP4V2* gene affect lipid metabolism in RPE cells, leading to RPE dysfunction and subsequent photoreceptor degeneration[9,10]. Mitochondrial damage may be the central node of RPE dysfunction[11]. Various degrees of atrophy in the photoreceptor interdigitation zone (IZ) and ellipsoid zone (EZ) bands are reported in BCD patients[12–15], which is an early sign of photoreceptor abnormality[2,16]. The *Cyp4v3* gene is the mouse ortholog of human *CYP4V2*. The proteins share 82% identity and 92% similarity[17] and the murine model can be used as an ideal disease model to explore BCD pathogenesis and therapy.

Recently, several teams including ourselves have reported encouraging results on *CYP4V2* gene replacement therapy in BCD-iPSC-RPE cells and *Cyp4v3* KO mice[18–20]. Our team has developed a *CYP4V2* gene replacement therapy drug ZVS101e[21], and two Investigator-Initiate Clinical Trials (IIT; NCT04722107 and NCT05714904) have been launched since June 2021. Up to now, a total of 16 participants have been enrolled, of which 6 have completed a one-year follow-up, and the results showed good safety and significant improvement in visual function[21]. From the previous report of Luxturna, visual improvement was maintained up to 7.5 years in human subjects[22]. However, loss of previously established transgene expression has also been reported in both animals and humans[23–25]. Nearly 80% of BCD patients carry mutations in exons 7 to 11 of *CYP4V2* gene, and three variants c.802-8_810del17insGC/GT, c.992A>C and c.1091-2A>G can account for more than 70% of the mutations identified[26]. In view of this, if we can find a suitable genome editing method for this mutation hotspot of *CYP4V2* gene, a single injection will achieve lifelong effectiveness.

Precise genome editing can be realized using the double-strand breaks (DSBs)-dependent CRISPR/Cas systems through two major pathways: homology-directed repair (HDR) or non-homologous end joining (NHEJ)[27]. Under the guidance of a single-guide RNA (sgRNA), CRISPR/Cas9 nuclease induces DNA DSBs at specific site and achieves precise knock-in in the presence of donor DNAs. The HDR pathway can realize site-specific transgene integration but is limited by low efficiency in non-dividing cells such as RPE cells[28]. Alternatively, the NHEJ-mediated knock-in method can achieve precise gene correction by homology-independent targeted integration (HITI) strategy[29]. This HITI-based method has been shown to improve vision by subretinal injection in a rat model of retinitis pigmentosa[29] and ameliorate the disease progress via intravenous administration in the Adrenokeukodystrophy mice[30]. HITI application in the liver, Duchenne Muscular Dystrophy, hemophilia B, and others has also been reported[31–37].

Here, we report that HITI-based editing can achieve precise DNA knock-in in BCD patient-derived iPSCs. Furthermore, we applied the HITI-based editing to a humanized *Cyp4v3* mouse model (*h-Cyp4v3*^*mut/mut*) and achieved targeted integration of exon 7-11 of human *CYP4V2* gene into the intron 6 by the delivery of two AAV vectors. Long-term treatment showed that HITI-based editing restores visual function and rescue photoreceptor and RPE degeneration in vivo. Our study suggested that HITI-based editing can be used as a suitable and reliable approach to develop gene therapies for BCD, and potentially for other IRDs diseases.

## Results

### In vitro genome editing using the HITI-based method

To examine whether the CRISPR/Cas9-mediated HITI-based method achieves efficient and precise knock-in in vitro, the sgRNAs and donors were screened and validated in HEK293T cells. A schematic of gene editing by HITI is shown in Fig.1a, the sgRNA guides Cas9 to generate double-strand cleavage on intron 6 of the *CYP4V2* gene, and the HITI donor insert the carried sequence, part of intron 6 and exon 7-11 into the break position to achieve precise repair. Among seven sgRNA plasmids (i.e. pX601-CMV-SaCas9-puro-sgRNA1-7) tested, sgRNA3 and

sgRNA4 had better cleavage efficiency (Fig.1b, c). Subsequently, the sgRNA3/4 plasmids and the matched donor plasmids pMD™19-T-donor3/4-EGFP were co-transfected into HEK293T cells, separately. Both sgRNA3/sgRNA4 and their corresponding donors achieved high fidelity insertion (Fig. 1d). The majority of insertions (76.7%) occurred at the 5 'junction site of sgRNA4 has a duplication of stub sgRNA fragment generated by cutting (stub duplication) as expected. Surprisingly, 78% of the sequences at the 5 'junction site of sgRNA3 were identical to wild-type (WT). The infidelity repair was caused by the indels of donor fragments.

Compared with the WT sequence, the HITI repaired sequence will produce small stub duplication at the cutting site in intron 6 of the *CYP4V2* gene, which may lead to abnormal splicing at the mRNA level and must be carefully evaluated in HITI genome editing therapy. Therefore, we used the minigene method to detect the possible aberrant splicing of repaired sequences. The negative control pMD™19-T-mgCYP4V2_exon6-intron6-exon7 (pmg_WT) will produce normal splicing, the positive control pMD™19-T-mgCYP4V2_exon6-c.802-8_810del17insGC-exon7 (pmg_mut) will result in abnormal splicing, the pMD™19-T-mgCYP4V2_exon6-sgRNA3 stub duplication-exon7 (pmg_sgRNA3) and pMD™19-T-mgCYP4V2_exon6-sgRNA4 stub duplication-exon 7 (pmg_sgRNA4) are experimental groups carrying the repaired sequence with stub duplication generated after editing (Supplementary Fig. 1a). We transfected the above four minigenes in HEK293T cells separately. RT-PCR and sequencing showed that pmg_sgRNA3/4 all produced normal splicing products (Fig. 1e, f and Supplementary Table 1). The results suggest that the stub duplications produced after HITI genome editing do not affect the normal mRNA splicing of the editing gene. Western blot results showed that the repaired sequence can be effectively translated (Fig.1g, Supplementary Fig. 1b). Combined with the editing efficiency results above, sgRNA3 and its corresponding donor were chosen for further study.

In addition to the editing efficiency, the off-target effect is another important factor to be addressed in CRISPR/Cas9-mediated gene editing therapy. Through Cas-OFFinder (www.rgenome.net/cas-offinder/)[38], all potential off-targets with no more than 3 mismatches to positions 4-20 of sgRNA3 with/without no more than one bulge were selected in the genome. No target with less than 3 mismatches or 1 bulge was found. Twenty-four targets with 3 mismatches and 1 bulge considered at high off-target risk are verified with no detectable off-target activity through Next-generation sequencing (Supplementary Table 2). T7E1 assay and sequencing analysis of these targets failed to find off-target sites. We also examined possible off-target sites at the genome-wide level using GUIDE-seq[39] and found no off-target sites (Fig.1h). The above results demonstrated that the cleavage activity of sgRNA3 was site-specific.

### Application of HITI-based editing to BCD patient-derived iPSCs

One BCD patient and one healthy control (WT) were recruited. The patient manifested characteristic changes of BCD, with yellow-white tiny crystalline deposits scattered bilaterally, spectral domain-optical coherence tomography (SD-OCT) showed hyper-reflective pre-epithelial deposits and disturbed organization or even loss of the RPE-photoreceptor outer/inner segment layer. The patient carried a homozygous mutation of c.802-8_810del17insGC, resulting in the loss of exon 7 in *CYP4V2* transcript (Fig. 2a). The urine cells (UCs) collected from the BCD patient and WT were reprogrammed into iPSCs (Fig. 2b). They exhibited typical iPSC characteristics with a high nuclear-to-cytoplasmic ratio, expressed pluripotency markers NANOG, OCT4, SSEA4 and TRA-1 (Fig. 2c) and showed differentiation potential marked by AFP, SMA and TUJ1 (Supplementary Fig. 1c), without significant difference between BCD and WT.

We further investigated the feasibility of HITI-based genome editing in BCD-iPSCs. The pX601-CMV-SaCas9-puro-sgRNA3 and

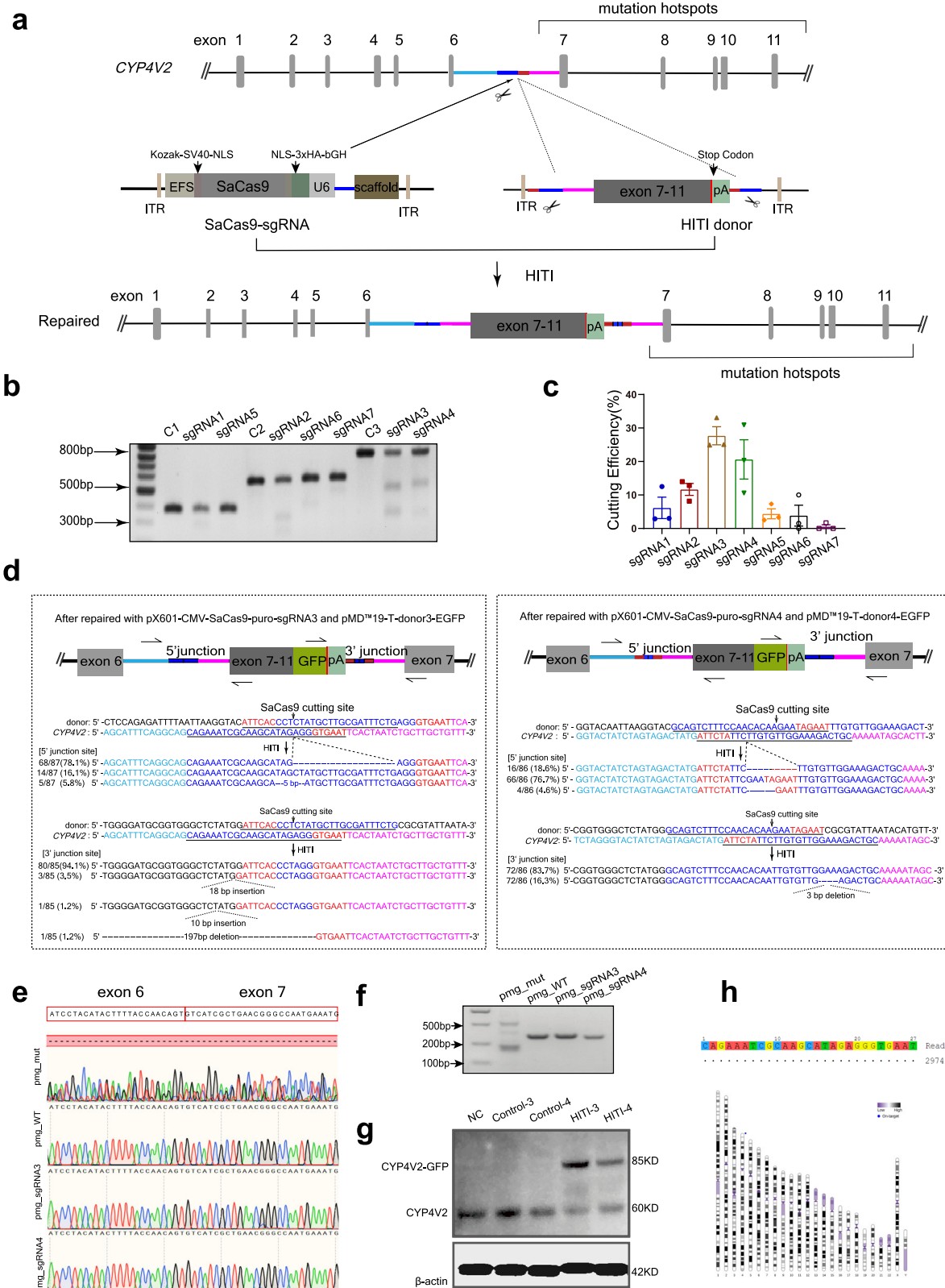

pX601-CMV-Puro-P2A-mCherry-donor3 (Fig. 2d) were co-transfected into BCD-iPSCs. Three homozygous monoclonal cell clones (Clones 1-13, 8-13, and 21-14) with a single editing band were obtained (Supplementary Fig. 1d) after repeated screening using two pairs of PCR primers (Supplementary Table 1). Sequencing results showed that they all carried precise genome editing (Fig. 2d), and the repaired *CYP4V2* normally transcribed and translated (Fig. 2e, f, Supplementary Table 1).

**HITI-based editing restored the vitality and function of iPSCs-RPE cells derived from BCD patients**

In order to further verify whether HITI-based genome editing can restore the function of the mutant protein, five iPSC cell lines, WT-iPSCs, BCD-iPSCs, and three edited BCD-iPSCs cell clones were established and differentiated into NR-RPE organoids, from which RPE cells were isolated, identified and evaluated. They all possess the

**Fig. 1 | In vitro genome editing via CRISPR/Cas9-mediated HITI-based targeted integration. a** Schematic overview of HITI-based gene editing at intron 6 of *CYP4V2* gene. The Aqua rectangle indicates the sequence before the target on intron 6. The blue rectangle indicates SaCas9 crRNA sequences. The red rectangle indicates the PAM sequence. The plum rectangle indicates the sequence after the target on intron 6. **b** Seven sgRNAs cutting efficiency study with T7E1, sgRNA3/4 with higher cutting efficiency were selected. **c** Quantification of T7E1 cutting efficiency, *n* = 3 biological replicates. Error bars represent the mean ± SEM. **d** Schematic and sequencing analysis of 5′ and 3′ ends of the integration sites of *CYP4V2* after co-transfection with pX601-CMV-SaCas9-puro-sgRNA3/4 and the matched donor plasmids pMD™19-T-donor3/4-EGFP in HEK293T cells. The arrows indicate the primers used to amplify the junction flanked on endogenous *CYP4V2* and the HITI donor after integration. EGFP is shown in green. **e, f** Minigene was used to detect the possible aberrant splicing of repaired sequences using Sanger sequencing (**e**) and RT-PCR (**f**), and results showed that pmg_sgRNA3/4 produced normal splicing products as pmg_WT. **g** Western blot results showed that the repaired sequence can be effectively transcribed and translated. CYP4V2-GFP was detected with anti-CYP4V2 antibody, indicating repaired CYP4V2. NC, HEK293T cells without transfection; Control-3/4, cells transfected with pMD™19-T-donor3/4-EGFP; HITI-3/4, cells co-transfected with sgRNA3/4 plasmid and matched donor plasmid pMD™19-T-donor3/4-EGFP. **h** Off-target examination of sgRNA3 by genome-wide, unbiased identification of DSBs enabled by sequencing (GUIDE-seq), no off-target sites were detected. Source data are provided as a Source Data file.

morphological characteristics of native RPE cells, including polygonal and pigmented with positive expression of markers CRALBP and PAX6 (Fig. 2g). The CCK8 assay showed that compared with WT-derived iPSC-RPE, the viability of iPSC-RPE cells from BCD patients was significantly reduced. In contrast, the cell viability of the RPE derived from edited BCD-iPSCs was significantly improved again (Fig. 2h).

## Generation and characterization of a humanized *Cyp4v3* mouse with mutations

We generated a novel humanized knock-in *Cyp4v3* mouse (*h-Cyp4v3^{mut/mut}*) by CRISPR/Cas9-mediated HDR. In this *h-Cyp4v3^{mut/mut}* mouse model, the m*Cyp4v3* genomic DNA from exon 6 (mE6) to exon 8 (mE8) was replaced with the human counterpart containing two mutations: E7: c.802-8_810del17insGC and E8: c.992A>C. These mutations resulted in the deletion of exon 7 and generation of the mutation p.H331P (Fig. 3a). This DNA replacement in the *h-Cyp4v3^{mut/mut}* mouse was confirmed by DNA sequencing and the expected exon 7 deletion was further verified with RT-PCR and Western blot (Fig. 3b, c, Supplementary Table 1). Commonly encountered retinal degeneration mutations (*Pde6b^{rd1}*, *Crb1^{rd8}*, *Pde6b^{rd10}*, and *Rpe65^{rd12}*) were excluded in the mouse model using Sanger sequencing (Supplementary Table 1). Forty potential off-target sites through Cas-OFFinder were verified with no detectable off-target activity using sequencing analysis (Supplementary Table 3). Moreover, *h-Cyp4v3^{mut/mut}* were born with the expected Mendelian ratios and were healthy and fertile. The above experiments clearly showed that the *h-Cyp4v3^{mut/mut}* mouse carried two mutations (E7: c.802-8_810del17insGC and E8: c.992A>C) was established successfully.

In the *h-Cyp4v3^{mut/mut}* mouse, scattered yellow-white granular deposits were found at 6 months. These deposits increased in number and became larger and confused with aging (Fig. 3d). The yellow-white granular deposits present as hyper-reflective foci on OCT (Fig. 3e). Interestingly, hypo-reflective lesions between the EZ and IZ were identified in *h-Cyp4v3^{mut/mut}* mouse, which were remarkable when compared to WT group at 12 months old.

Concomitant with morphological changes in fundus, the amplitude of ERG also exhibited abnormalities. At the age of 6 months, the amplitude of the dark-adapted b-wave of the *h-Cyp4v3^{mut/mut}* mice reduced at 0.3 and 1.0 cd*s/$m^2$ scotopic stimulus intensities compared to WT mice. At the age of 9 months, the reduction extended to 0.03, 0.1, 0.3, and 1.0 cd *s/$m^2$ scotopic stimulus intensities. It continued to decline with aging, and all seven scotopic stimulus intensities were involved at 12 months. Different from the scotopic condition, the amplitude of light-adapted b-wave started to decrease from 12 months (Fig. 3f, g).

Based on the previous results that a significant phenotypic difference between *h-Cyp4v3^{mut/mut}* mice and WT mice was observed at 12 months of age, we examined mice at this age point to further investigate the morphology and metabolism changes in RPE and photoreceptors in *h-Cyp4v3^{mut/mut}* mice. For RPE cells, phalloidin staining showed that the boundary of RPE cells was blurred and exhibited 'railroad tracks'-like arrangement (Fig. 4a)[40]. TEM examination showed accumulation of lipid droplets (LDs) in RPE cells,

excessive metabolic residues between RPE cells, and the increased, disordered, and crimped basal membrane. For photoreceptors, abnormal IZ and EZ bands were observed. Mitochondria in IZ band reduced in lamellar cristae, and mitochondria in EZ band showed blurry of lamellar cristae and swollen intercristal spaces with vacuole (Fig. 4b). Quantification of the membrane structure profile revealed that the mitochondrial diameter increased significantly (Fig. 4c), and the number of mitochondria was significantly reduced (Fig. 4d). A disorder in the outer nuclear layer (ONL) layer and deposition were found at 250 μm away from the optic nerve head (ONH) (Fig. 4e). The number of photoreceptor nucleus per field (50 μm × 50 μm) significantly reduced, while the difference in thickness is not obvious (Fig. 4f); Photoreceptors exhibited significant nuclear disorder and atrophy, with widespread vacuoles in the ONL region (Fig. 4g).

In addition to the morphological observation, we further analyzed the changes in lipid metabolism through LC−MS/MS analysis. Compared with WT mice, the value of fifteen free fatty acids (FFAs) were higher in *h-Cyp4v3^{mut/mut}* mice, including FFA22:0 (*p* = 0.0415)、FFA22:1(*p* = 0.0428)、FFA23:0 (*p* = 0.0256)、FFA24:0 (*p* = 0.0338)、FFA25:0 (*p* = 0.0268)、FFA25:1(*p* = 5.38e-5) and FFA26:0 (*p* = 0.00754). On the other hand, the other three types of FFA (FFA20:2, FFA20:5, and FFA22:6.) are lower than WT.

## HITI-based method achieves precise genome editing in *h-Cyp4v3^{mut/mut}* mouse

We generated AAV vectors using sgRNA3 and the corresponding HITI donor based on their in vitro high precise repair efficiency. For the treatment group, rAAV2/8-EFS-SaCas9-sgRNA3 (abbreviated as rAAV2/8-SaCas9-sgRNA3) and rAAV2/8-EFS-Puro-P2A-mCherry-donor3 (abbreviated as rAAV2/8-HITI-donor3) were used (Fig. 5a). For empty vector group, rAAV2/8-EFS-SaCas9-blank (abbreviated as rAAV2/8-SaCas9-blank) and rAAV2/8-HITI-donor3 were used. We delivered dual viral vectors into the subretinal space of 1-month-old *h-Cyp4v3^{mut/mut}* mice. Each eye (*n* = 44) received 2 μL of co-delivered rAAV vectors mixed 1:1 at different concentrations. In the low-dose treatment group, the two viral vectors were $1 \times 10^9$ vg/μL respectively, with a total load of $2 \times 10^9$ vg. In the high-dose treatment group, the two viral vectors were $3 \times 10^9$ vg/μL respectively, with a total load of $6 \times 10^9$ vg. In the empty vector group, the two viral vectors were $1 \times 10^9$ vg/μL respectively, with a total load of $2 \times 10^9$ vg. One month post-injection, flat mounts and IF analysis of full-view C sections results showed that mCherry fluorescence expressed widely in the retina (Supplementary Fig. 2a, b). Higher power images of retinal sections indicated that the EFS promoter drives strong gene expression in RPE cells as well as the outer and inner segment (OS/IS) and outer nuclear layer (ONL) of photoreceptor cells (Supplementary Fig. 2c). Quantitative PCR confirmed efficient Cas9 mRNA and gRNA expression in two doses groups (Supplementary Fig. 2d, e, Supplementary Table 1) at 5 months after treatment, Western blot further revealed the stabilization of Cas9 protein in mouse retinal-RPE complexes (Supplementary Fig. 2f). Retinal-RPE complexes of posttreatment mouse were taken to measure proportion of edited mRNA transcripts at 2, 5, 8, 11 months after treatment (i.e., 3, 6, 9,

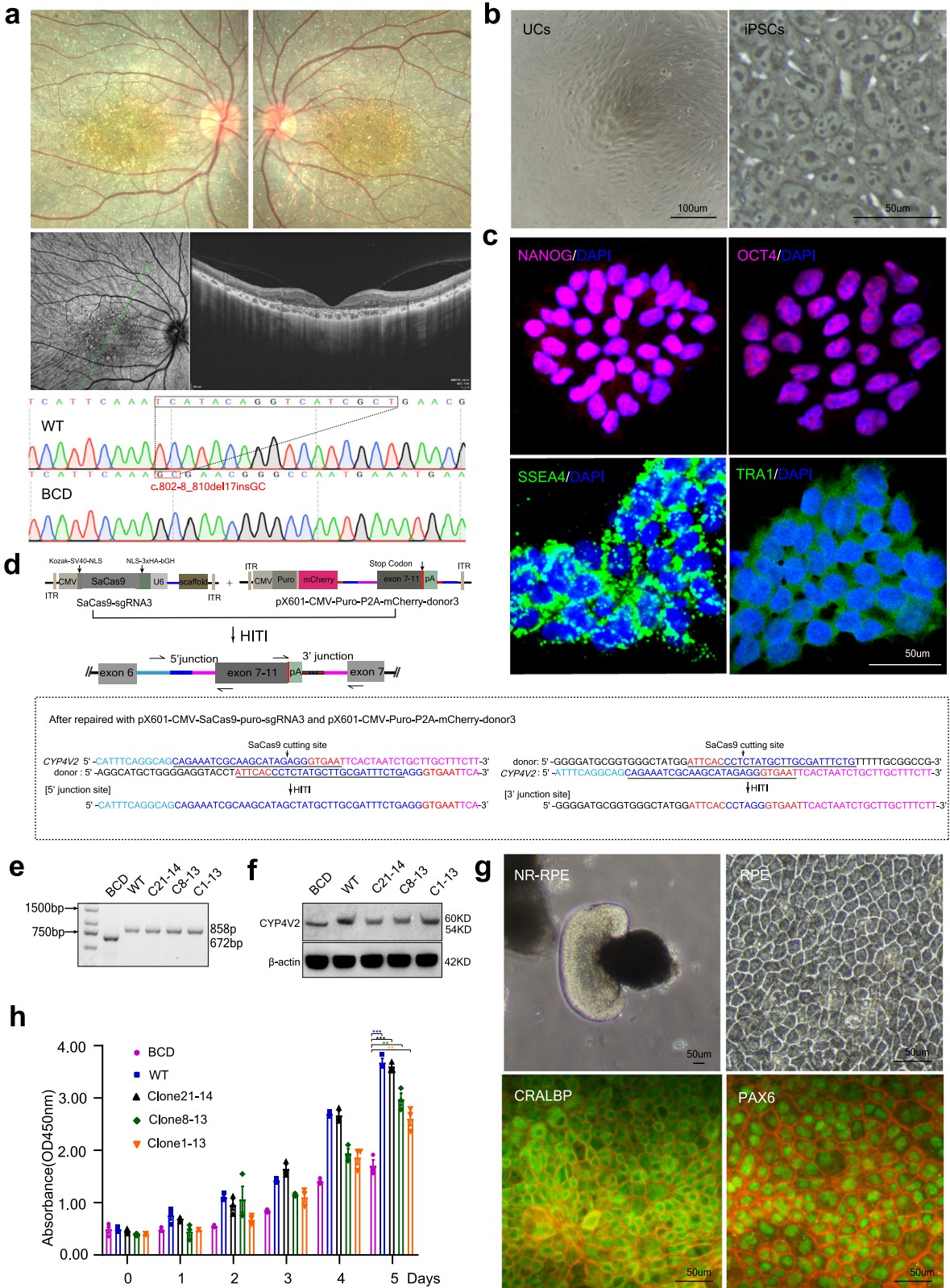

12 months after birth). Quantification of both edited and unedited cDNA was measured using real-time qPCR with two primer pairs binding to hE10, hE11(edited sequence) and mE9, mE10(unedited sequence) respectively (Supplementary Table 1). The proportion of edited transcripts is calculated using the following formula: quantity of edited cDNA/(quantity of edited cDNA+ quantity of unedited cDNA). In order to quantify the potential transcript contamination transcribed from uninserted donor, we conducted the above calculation in the empty vector group ($n = 4$, injected with donor only). The displayed value does not exceed 0.1% and can be ignored. In the low-dose group, the proportion of edited transcripts at 2, 5, 8, and 11 months after treatment was 3.0% ± 3.3%, 18.6% ± 16.3%, 25.3% ± 15.5%, and

**Fig. 2 | Precise *CYP4V2* repair in BCD patient-derived iPSC cells using HITI. a** The patient manifested characteristic changes of BCD, yellow-white tiny crystalline deposits scattered bilaterally, spectral domain-optical coherence tomography (SD-OCT) showed hyper-reflective pre-epithelial deposits and disturbed organization or even loss of the RPE-photoreceptor outer/inner segment layer. Genetic analysis showed the patient carried a homozygous mutation of c.802-8_810del17insGC. **b** Urine cells (UCs) isolated from BCD patient, Scale bar = 100 μm; iPSCs colony emerged after retroviral transduction, showing typical characteristics with a high nuclear-to-cytoplasmic ratio, Scale bar = 50 μm. **c** Immunofluorescence staining of NANOG, OCT4, SSEA4 and TRA-1 in iPSCs colony. Scale bar = 50 μm. **d** Schematic and sequencing analysis of 5′ and 3′ ends of the integration sites of *CYP4V2* after repair with pX601-CMV-SaCas9-puro-sgRNAs3 and pX601-CMV-Puro-P2A-mCherry-donor3 in BCD-iPSC cells. The arrows indicate the primers used to amplify the junction flanked on endogenous *CYP4V2* and the HITI donor after integration.

The blue rectangle indicates SaCas9 crRNA sequences. The red rectangle indicates the PAM sequence. **e** RT-PCR genotype analysis showed that C21-14, C8-13 and C1-13 achieved precise genome editing. **f** Western blot results showed that the repaired sequence can be effectively transcribed and translated. **g** NR-RPE organoids and the detached RPE cells. Immunofluorescence staining showed they have a positive expression of RPE markers CRALBP and PAX6. Scale bar = 50 μm. **h** The CCK8 assay was used to compare the cell viability of iPSC-RPE. The results showed that compared with WT-iPSC-RPE, the viability of BCD-iPSC-RPE cells was significantly reduced, and it was significantly improved after HITI therapy. Student's unpaired two-tailed t-test. Error bars represent the mean ± SEM. $n = 3$ biological replicates. BCD vs. WT, $p = 0.000116$; BCD vs. Clone21-14, $p = 8.5e-5$; BCD vs. Clone8-13, $p = 0.00134$; BCD vs. Clone1-13, $p = 0.00591$. **$p < 0.01$, ***$p < 0.001$. Source data are provided as a Source Data file.

29.3% ± 12.2% (mean ± SEM). The proportion in the high-dose group was 11.0% ± 9.7%, 29.3% ± 2.1%, 33.9% ± 19.2%, and 31.9% ± 9.7% (mean ± SEM), respectively (Fig. 5b). The ratio of edited transcripts increased with treatment time, reached a plateau (about 30%) at 8 months after treatment, and maintained stability to 11 months. At 2 and 5 months after treatment, the high-dose group performed better than the low-dose group, but no significant at 11 months after treatment.

We used DNA cloning and Sanger sequencing to identify the sequences of the 5′ and 3′ junction sites carrying insertion fragments. Each group has 4 eyes, with ≥ 50 clones per eye. We found that at the 5′ end the proportion of fidelity editing after 2, 5, 8, and 11 months of treatment was 88.3% ± 2.7%, 83.3% ± 3.1%, 84.5% ± 2.6%, and 84.1% ± 5.4% in the low-dose group, 89.8% ± 1.1%, 84.7% ± 6.1%, 86.0% ± 2.8%, and 84.8% ± 4.2% in the high-dose group. At the 3′ end, the ratios are 93.4% ± 3.9%, 91.7% ± 2.2%, 87.0% ± 2.5%, and 88.1% ± 4.6% in the low-dose group and 90.3% ± 2.3%, 92.8% ± 4.2%, 84.0% ± 4.3%, and 86.1% ± 2.6% (mean ± SEM) in the high-dose group (Fig. 5c). There is no significant difference between dose groups and among treatment times. Pooled two doses cloning sequences revealed that 11.26% (5′ junction) and 7.5% (3′ junction) were infidelity repair with rAAV inverted terminal repeats sequence (ITRs) insertion or bases deletion/insertion in the stub duplication region, 1.7% (5′ junction) and 3.17% (3′ junction) were infidelity repair with partial donor repeat insertion/deletion in the flanked sequence. (Fig. 5d). Those sequences without carrying insertion fragments were undetected using cloning. Therefore, we performed a PCR-based next-generation sequence (NGS) analysis in the high-dose group at 11 months after treatment (Supplementary Fig. 3a). AUGUSTUS predicted that all sequences with/without carrying insertion fragment introduced repaired/unrepaired transcripts, which were translated to repaired/mutant protein, except for infidelity repair with large fragment deletion at 3′ region junction (4/2876) (Supplementary Fig. 3b). BioEdit revealed the corresponding amino acid sequences of the non-functional proteins translated from those 4 abnormal transcripts (Supplementary Fig. 3c). Apart from that, we also verified transcripts using paired primer flanked on mE2 and hE9 to amplify the repaired transcript sequence. The result showed that only the treatment group showed a positive amplified band while no band was acquired in the WT and empty vector group (Fig. 5e and Supplementary Table 1). Sanger sequence further confirmed the transcript sequence (Supplementary Table 1). Western blot demonstrated that two blots were shown in the treated group, the shorter one is in line with the truncated protein from *h-Cyp4v3*^mut/mut^ mice, and the larger one is in line with the blot from NIH3T3 cells transfected with plasmid pAV-CAG-CYP4V3(E1-6)-CYP4V2(E7-11) containing the desired edited sequence (Fig. 5f). IF analysis of retinal sections showed Cyp4v3 staining in the RPE cells (Supplementary Fig. 3d). The above data demonstrate that HITI-based editing therapy not only corrects the mutation, but also restore normal transcription and translation in mice.

## HITI-based method restores visual function in *h-Cyp4v3*^mut/mut^ mouse

To determine if reduced ERG in *h-Cyp4v3*^mut/mut^ mice can be restored, ERG responses were examined at 2, 5, 8, and 11 months after treatment. To avoid interference from surgical procedures, we first compared the ERG between *h-Cyp4v3*^mut/mut^ mice without operation and the empty vector group. At 2 months after treatment, the empty vector group showed a significant decrease in b-wave amplitude of scotopic stimulation intensity at 3 cds/m² ($p = 0.00617$), 10 cds/m² ($p = 0.000311$), and photopic stimulation intensity at 10 cds/m² ($p = 0.0498$) and 30 cds/m² ($p = 0.0238$), when compared to *h-Cyp4v3*^mut/mu^ mice without operation. Since from 5 months after treatment, the above differences were no longer statistically significant, indicating that the influence caused by operation can be regarded as negligible after 5 months (Supplementary Fig. 4a). When compared with *h-Cyp4v3*^mut/mut^ mice at the same age, both low-dose and high-dose groups showed varying degrees of decreases in ERG amplitude at 2 months after treatment. At 5 months after treatment, the ERG amplitude of the low-dose treatment group was slightly better than that of the *h-Cyp4v3*^mut/mut^ mice, especially the b-wave amplitude under 0.3 cd*s/m² scotopic stimulus intensities ($p = 0.0454$), while the ERG amplitude of the high-dose treatment group was slightly worse than that of the *h-Cyp4v3*^mut/mut^ mice, especially the b-wave amplitude under 10 cd*s/m² scotopic stimulus intensities ($p = 0.00799$). At 8 months after treatment, the b-wave amplitudes of the two-dose treatment groups increased significantly under most scotopic stimulus intensities, with no significant difference between the low and high-dose groups. At 11 months after treatment, the high-dose treatment group showed a significant increase in b-wave amplitude under 7 scotopic stimulus intensities, and the effect was more significant than the low-dose treatment group. In addition, at this time point, the high-dose treatment group mice also showed a significant increase in b-wave under 10 cd*s/m²($p = 0.0312$) and 30 cd*s/m² ($p = 0.00605$) bright stimulus intensities (Fig. 5g). Representative diagrams of scotopic/photopic b-wave were shown in Fig. 5h at 11 months after treatment and other time points (Supplementary Fig. 4b). The results revealed that both low- and high-dose groups showed significant therapeutic effects after subretinal injection. The low-dose group had a more significant effect at 5 months after treatment, while the high-dose group had a more significant effect at 11 months after treatment.

## HITI-based method rescue photoreceptor and RPE degeneration in *h-Cyp4v3*^mut/mut^ mouse

We selected mice at 11 months after treatment to investigate the changes in RPE cells and photoreceptors. Compared with the empty vector group at the same age, two doses treatment groups performed significant reduction of yellow-white deposition (Fig. 6a) and hypo-reflective lesion between the IZ and EZ bands (Fig. 6b). Similarly, phalloidin staining showed that the boundaries of RPE cells became clear with normal morphological features (Fig. 6c). The TEM revealed

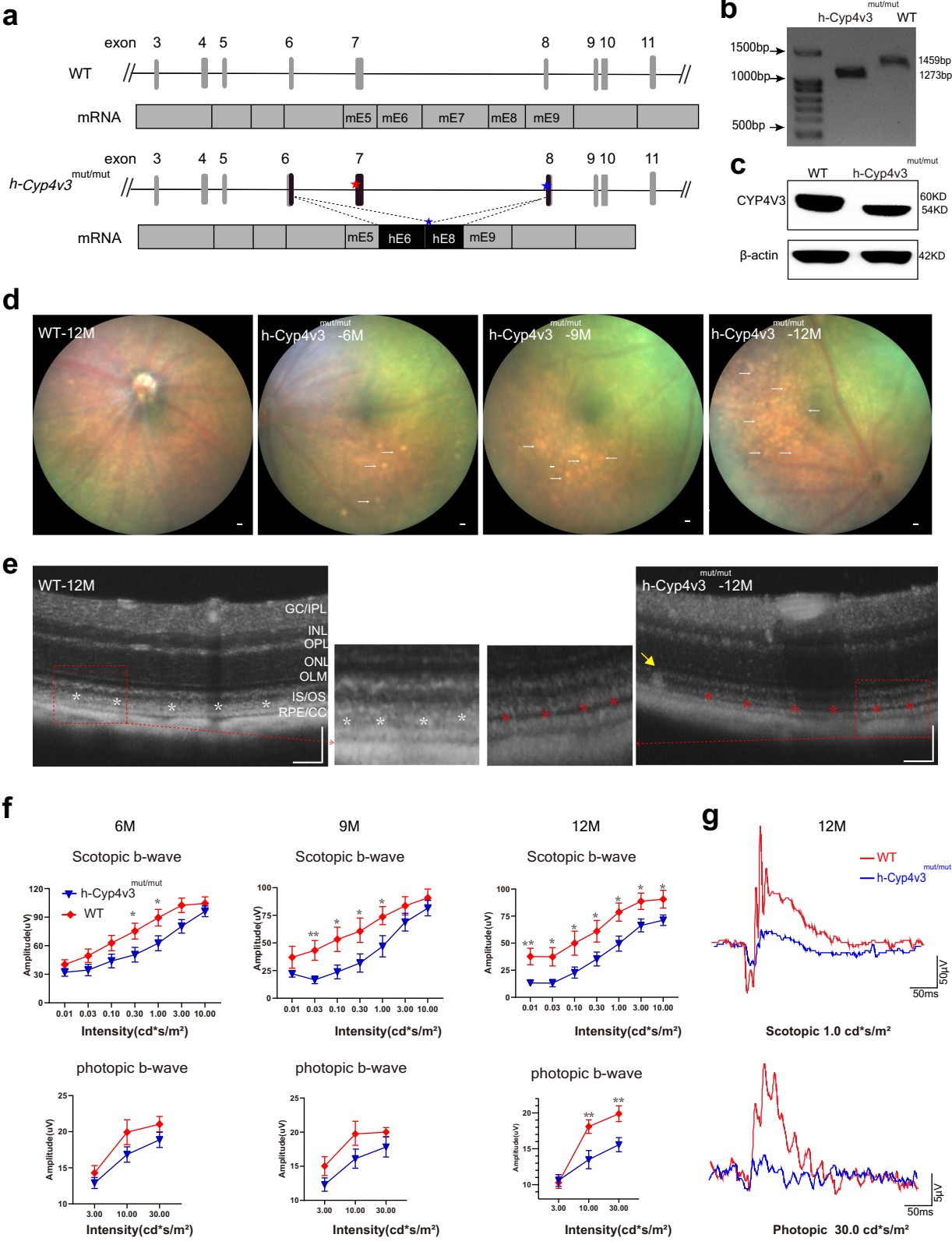

almost no accumulation of lipid droplets (LDs) and deposition of metabolic residues in RPE cells after treatment (Fig. 6d). For photoreceptors, HE and TEM showed that the number of photoreceptors increased (Fig. 6e–g), accompanied with the decrease of swelling mitochondria diameter and increase of mitochondria number and lamellar cristae in the IZ and EZ bands (Fig. 6h and Supplementary Fig. 4c).

LC–MS/MS analysis of FFAs profiling of the RPE-choroid complex indicated that the upregulated FFAs in *h-Cyp4v3*^{mut/mut} mice were decreased after treatment. Correspondingly, FFAs downregulated in *h-Cyp4v3*^{mut/mut} mice were upregulated after treatment (Fig. 6i). The above results clearly show that HITI genome editing therapy has therapeutic effects on the improvement of the morphology, number, and metabolism of RPE and photoreceptors in the retina.

**Fig. 3 | Generation and characterization of a humanized *Cyp4v3* mouse containing two mutations, E7: c.802-8_810del17insGC and E8: c.992A>C (*h-Cyp4v3^mut/mut^*). a** Schematic diagram of DNA and mRNA structures in wild-type (WT) and *h-Cyp4v3^mut/mut^* mice. The red star indicates mutation of E7: c.802-8_810del17insGC; the Blue star indicates mutation of E8: c.992A>C. b-c RT-PCR (**b**) and Western blot (**c**) results confirmed that the resulting sequence is shorter than WT. **d** Representative fundus images from WT and *h-Cyp4v3^mut/mut^* mice, scattered yellow-white granular deposits were found in *h-Cyp4v3^mut/mut^* mice. Scale bars:50 μm. **e** Representative OCT images from WT and *h-Cyp4v3^mut/mut^* mice. A hypo-reflective lesions between the photoreceptor EZ and IZ were identified in *h-Cyp4v3^mut/mut^* mouse, which was indicated with a red star. The normal structure was indicated with a white star. Lipid deposition was indicated with a yellow arrow. GC/IPL ganglion cell/inner plexiform layer, INL Inner Nuclear Layer, OPL Outer Plexiform Layer, ONL Outer Nuclear Layer, OLM Outer Limiting Membrane, IS/OS Inner segment/Outer segment, RPE/CC Retinal Pigment Epithelium/Choriocapillaris.

Scale bars:50 μm. **f** Statistic graph of ERG amplitude of scotopic/photopic b-wave in 6, 9 and 12 months-old WT ($n = 19, 8, 13$ eyes) and *h-Cyp4v3^mut/mut^* mice ($n = 34, 14, 11$ eyes). Student's unpaired two-tailed *t*-test. Error bars represent the mean ± SEM. Exact *P*- values: (0.30 cd*s/$^{m2}$ and 1.0 cd*s/$^{m2}$ scotopic stimulus intensities in 6 months) $p = 0.0447$, $p = 0.0346$; (0.03 cd*s/$^{m2}$, 0.10 cd*s/$^{m2}$, 0.30 cd*s/$^{m2}$ and 1.0 cd*s/$^{m2}$ scotopic stimulus intensities in 9 months) $p = 0.00362$, $p = 0.0200$, $p = 0.0248$, $p = 0.0454$; (0.01 cd*s/$^{m2}$, 0.03 cd*s/$^{m2}$, 0.10 cd*s/$^{m2}$, 0.30 cd*s/$^{m2}$, 1.0 cd*s/$^{m2}$, 3.0 cd*s/$^{m2}$ and 10.0 cd*s/$^{m2}$ scotopic stimulus intensities in 12 months) $p = 0.00523$, $p = 0.0120$, $p = 0.0482$, $p = 0.0479$, $p = 0.0146$, $p = 0.0492$, $p = 0.0439$; (10.0 cd*s/$^{m2}$ and 30.0 cd*s/$^{m2}$ photopic stimulus intensities in 12 months) $p = 0.00811$, $p = 0.00468$. *$p < 0.05$, **$p < 0.01$. **g** Representative diagram of scotopic (intensity of 1.0 cd*s/m²) and photopic (intensity of 30.0 cd*s/m²) ERG waveforms of 12-month-old WT and *h-Cyp4v3^mut/mut^* mice. Source data are provided as a Source Data file.

## Discussion

In recent years, CRISPR/Cas9-based gene editing therapy for monogenic disease is rapidly emerging, and many have entered into clinical trials. In 2021, N Engl J Med reported the investigational use of CRISPR/Cas9-based in vitro gene editing (product CTX001) to treat Transfusion-dependent β-thalassemia and Sickle Cell Disease and completed phase III clinical trial two years later[41]. The first CRISPR/Cas9-based in vivo gene editing (product NTLA-2001) for Transthyroidosis Amyloidosis has also been reported subsequently[42]. Gene editing therapy opens a new era of medical treatment for genetic diseases. Different from gene replacement, gene editing therapy achieves lasting treatment by directly correcting mutations in the genome, and is considered to be a more promising method to manage genetic disorders. This study demonstrated that the CRISPR/Cas9-mediated HITI-based method can achieve precise and effective genome integration in the *CYP4V2* gene both in BCD-iPSCs and in *h-Cyp4v3^mut/mut^* mice, and this treatment can cover almost 80% of BCD mutations.

At present, CRISPR/Cas9-mediated specific genome integration can be achieved through the following four methods: NHEJ (mainly HITI), HDR, microhomology-mediated end joining (MMEJ), and homology-mediated end joining (HMEJ). HMEJ is reported to perform better than NHEJ, HDR, and MMEJ in cultured cells, animal embryos, and tissues in vivo[43], and even better than HITI in chicken primordial germ cells[44]. However, the editing efficiency of HMEJ in the retina in vivo was unknown. HITI has been verified to improve vision in the rat model of retinitis pigmentosa[29] and showed a promising future for retinal genetic disease.

Precise genomic correction is the most important consideration for successful gene editing-based drug development. Indels at the target site may adversely affect the HITI-based gene therapy, such as mRNA splicing of the edited gene. To identify these indels and evaluate their potential interference with the edited gene expression, we constructed minigenes that mimic stub duplications after repair and sequenced the DNA around the expected cleavage site. Minigene assays clearly showed that stub duplications did not interfere with mRNA splicing. Our results showed that the majority (95%) were fidelity integration in HEK293T cells and the ratio is 85% in *h-Cyp4v3^mut/mut^* mice, with no significant correlation with treatment time and dose. Most infidelity editing was the insertion of rAAV-ITRs at the integration site, which was consistent with previous reports that rAAV vectors can integrate at nuclease-induced DSBs[45–47]. When rAAV enters into the host cell, its single-strand genome is converted into double-strand linear DNA, the linear ITRs may be a more preferable source of DNA for integration into the DSBs than circular plasmids[46]. Alternatively, partially packaged rAAV may be enriched with ITRs, leading to more abundant ITRs for integration into the DSBs. This may explain why the precise integration in HEK293 cells is higher than that in vivo (95% vs. 85%). In this study, almost all sequences with/without carrying

insertion fragments were predicted have no additional protein product. Only four large deletions occurring in the 3′ region introduced abnormal transcripts and non-functional protein production (4/2876). Therefore, indels at 5′ and 3′ junction would likely have a negligible impact on the edited gene. Optimization of AAV vectors or development of non-viral vectors such as nanoparticles and liposomes will reduce this risk and further improve the safety of gene therapy drugs.

The editing efficiency is another key factor for clinical application. Based on the HITI method, we achieved about 30% edited transcripts in *h-Cyp4v3^mut/mut^* mice, higher than that reported 5% editing efficiency in *MERTK* rats[29,36]. Several factors contribute to the high proportion here. Firstly, related to the sgRNA, in vitro results showed that there are huge differences among different sgRNAs. Secondly, related to the treatment dose, the high-dose group performed better than the low-dose group in the long term. The third factor is related to the treatment time, the proportion of editing increased with treatment time, reaching a plateau after 8 months of treatment. The fourth factor is related to the size of the donor, the insertion fragment in this study was 30% smaller than that in the *MERTK* rat experiment. Though highly effective virus infection was achieved in the retina, the on-target assessments would be more accurate using dual viral vectors transduced cells rather than pan-retinal tissue. When testing the edited transcripts, we set the mRNA sample without reverse transcriptase as the control to exclude interference of residual DNA. However, there are still limitations. Such as limited sample size and differences within the group, which may also be related to surgical procedures. From previous reports, approximately 10% functional photoreceptors in the fovea are sufficient to achieve near-normal vision[48,49], we believe that this HITI editing treatment has potential clinical prospects. More samples and studies will obtain more reliable data in the future.

Off-target is another inevitable consideration in gene editing therapy. To accurately estimate off-targets, we excluded the use of mouse tissue and patient-derived iPSCs which may lead to underestimation and deviation. We chose HEK293T and performed biased (Cas-OFFinder) and unbiased (GUID-seq) detection, no high-risk off-target sites were found. The focus of this study is to evaluate the efficiency of the HITI method and test the feasibility of dual rAAV vectors therapy in vivo, more preclinical safety and toxicity studies were needed before clinical application. Both surgical operation and dosage affect the effectiveness evaluation[50]. Our results showed that subretinal injection caused retinal detachment, which has a big influence on visual function evaluation, it is consistent with our clinical results of BCD gene replacement therapy[21]. The fluid was gradually absorbed, the influence caused by the operation is no longer statistically significant after 5 months. In this study, the visual function of the high-dose group was worse than that of the low-dose group at 5 months after treatment, while the opposite was observed at 11 months after treatment. The possible reason is that high-dose vector administration leads to an overload of rAAV capsid protein and carrier

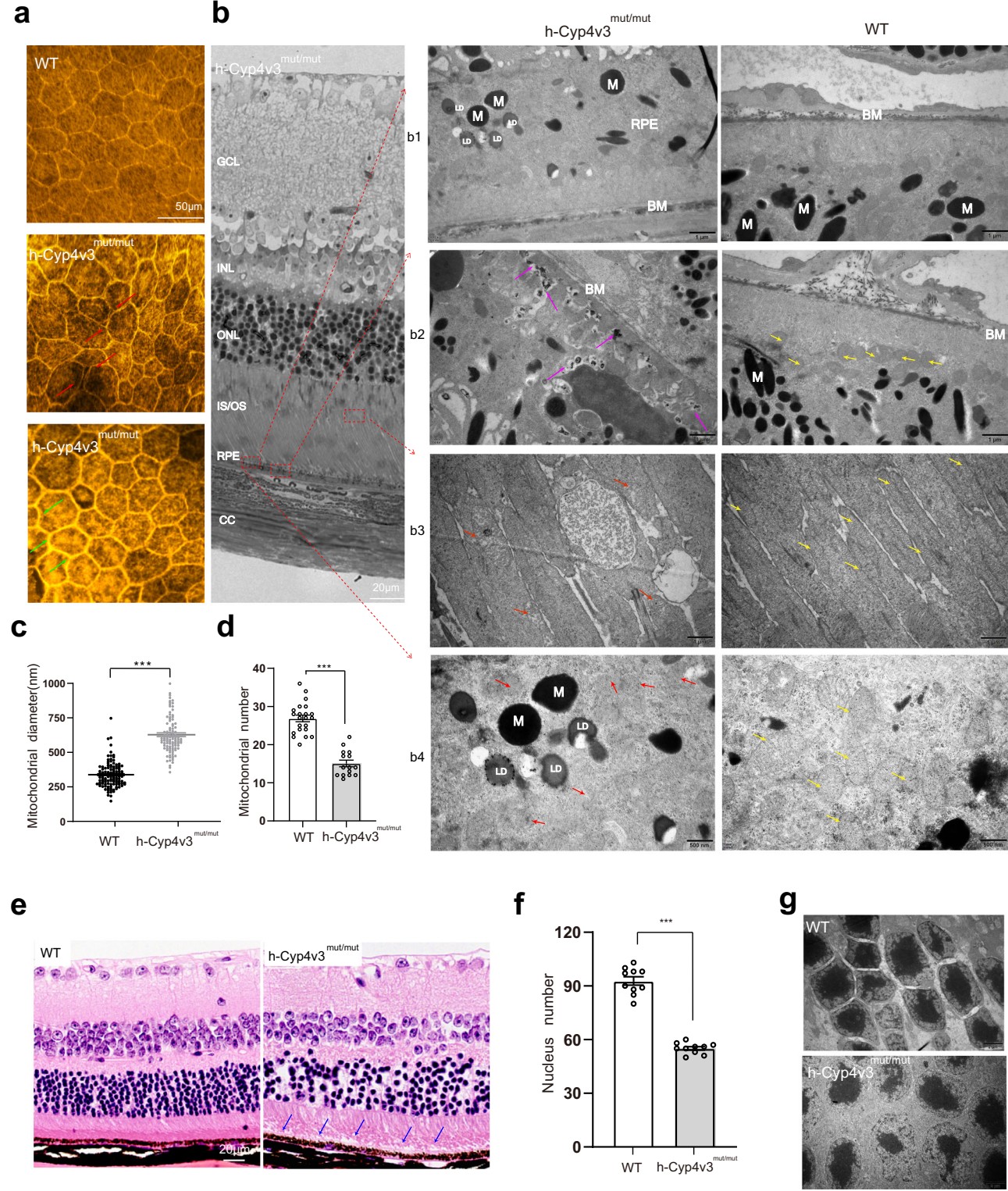

DNA, thereby exceeding the processing capacity of retinal cells in the short term[51], and after 8 months of treatment, the stress may be counteracted through metabolism. The results suggest that efficacy evaluation in clinical trials takes at least six months to be reliable.

To evaluate the efficacy of AAV vectors, we established the humanized BCD mice carrying two mutation hotspots (c.802-8_810del17insGC and c.992A>C) and used it to examine if visual abnormalities are recovered after drug administration. Compared with previous BCD knockout models[19,52,53], the *h-Cyp4v3^{mut/mut}* mice better

mimicked the clinical features of BCD patients. For instance, in addition to the commonly observed abnormalities such as yellow-white fundus deposits, ERG amplitude reduction, photoreceptor cell degeneration, and RPE damage, atrophy of IZ and EZ bands on OCT structure was clearly visible in the *h-Cyp4v3^{mut/mut}* mice. Subretinal administration of AAV vectors restored all the above abnormalities in *h-Cyp4v3^{mut/mut}* mice. The abnormal lipid metabolism, a proposed pathological mechanism for BCD[18,54] was detected in clinical manifestations (Fig. 3d–g), tissue sections (Fig. 4), molecular regulation

**Fig. 4 | Comparison of RPE and photoreceptor damage between 12-month-old *h-Cyp4v3*[mut/mut] mice and age-matched WT mice. a** Phalloidin staining of RPE flat mounts of 12-month-old WT and *h-Cyp4v3*[mut/mut] mice. Red arrowheads indicate the 'railroad tracks'-like arrangement of adjacent RPE cells; Green arrowheads indicate the blurred boundaries of RPE cytoskeleton. Scale bar = 50 μm. **b** Semi-thin (500 nm) light microscope photos (left) and transmission (70 nm) electron microscopy (TEM) retinal images (right) of *h-Cyp4v3*[mut/mut] and WT mice. Scale bar = 20 μm. Accumulation of lipid droplets (LDs) in RPE cells (b1). Excessive metabolic residues deposition between RPE cells and basal membrane indicated with purple arrowheads (b2). Abnormal mitochondria in EZ (b3) and IZ (b4) bands of photoreceptor with decreased and blurry lamellar cristae and vacuole, indicated with red arrowheads. Yellow arrowheads indicate regular mitochondria. CC choriocapillaris, RPE retinal pigment epithelium, IS/OS inner segments/outer segments, ONL outer nuclear layer, INL inner nuclear layer, GCL ganglion cell layer, M melanin pigment granule, BM basement membrane, LD Lipid droplets, N nucleus. Scale bars: 1 μm (b1-3); Scale bars: 500 μm (b4). **c, d** Quantification of the mitochondrial diameter (**c**) and numbers (**d**) in photoreceptor cells of WT and *h-Cyp4v3*[mut/mut] mice. $N = 119,108$ mitochondria were used for mitochondrial diameter analysis in WT and *h-Cyp4v3*[mut/mut] groups. Twenty-two (WT) and fifteen (*h-Cyp4v3*[mut/mut]) slices were used for analyzing the mitochondrial numbers. Student's unpaired two-tailed *t*-test. Error bars represent the mean ± SEM. **c** $p = 5.785e-49$, **d** $p = 2e-11$. ***$p < 0.001$. **e** Representative H&E retinal section images. The blue arrowheads indicate the deposition between RPE and photoreceptor cells. **f** Statistical diagram of the number of photoreceptors nucleus in the ONL per field (50 μm × 50 μm). Fields are taken from 250 μm away from the ONH in the retinal section. Student's unpaired two-tailed *t*-test. Error bars represent the Mean ± SEM, $n = 10$ eye sections. $p = 1.6e-11$, ***$p < 0.001$. **g** Images of representative TEM of the ONL. Source data are provided as a Source Data file.

(Fig. 6i), and ameliorated after treatment. CYP4V2 protein is thought to maintain the fatty acid homeostasis of the retina[8] thereby promoting RPE cell survival[55]. In agreement with these reports, most long-chain and very-long-chain polyunsaturated fatty acids in the RPE-choroid complex of *h-Cyp4v3*[mut/mut] mice were higher than those of WT mice, and the levels of these fatty acids were significantly reduced after treatment.

Gene therapy for IRDs is mainly based on gene replacement therapy. The complementary DNA (cDNA) of the targeted gene is packaged into a vector and injected into the subretinal space. Clinical trials have found that it can effectively improve patients' visual function[56,57]. Luxturna is currently the only approved gene therapy product in the field of IRDs. In previous studies, our group has developed a gene replacement drug ZVS101e for BCD. In two IIT clinical trials (IIT; NCT04722107 and NCT05714904), 80% of the patients had improved visual acuity after treatment, and about 40% of the patients had improved visual acuity ≥15 letters after treatment[21]. However, there are still several controversial points in gene replacement therapy. First, it is difficult to precisely regulate the expression level of foreign genes. Second, there are great differences in long-term effectiveness among different reports[22–25]. Third, due to the capacity limitation of AAV, large genes such as *USH2A* and *EYS* is difficult to be packaged into AAV. Fourth, gene replacement therapy is not applicable to those genes that gain abnormal function after mutation. Gene editing therapy can achieve precise genome repair. The endogenous regulatory sequence can precisely control the expression, and one treatment will be effective for a lifetime. Those genes that have mutation hotspots, such as *CYP4V2*, can try gene editing therapy. Up to now, 18 clinical trials of CRISPR gene editing therapy are in progress, including spinal muscular atrophy and hemophilia[58] (www.clinicaltrials.gov). Gene editing therapy and gene replacement therapy are complementary to each other, which can cover and benefit more patients.

In summary, our study demonstrated that the CRISPR/Cas9-mediated HITI-based method achieves efficient and precise genome repair of the *CYP4V2* gene both in vitro and in vivo. Our results suggest that HITI-based editing has a promising application to those IRDs genes with mutation hotspots, and can serve as an effective complement to gene replacement therapy.

## Methods

### Patients
This study conformed to the tenets of the Declaration of Helsinki. All experiments involving the patient's urine were approved by the Peking University Third Hospital Medical Ethics Committee (No. 2021262). Written informed consent was obtained from all participants, and the ethics committee approved this consent procedure.

One BCD patient and one healthy volunteer were enrolled in this study, and they were recruited from the Department of Ophthalmology, Peking University Third Hospital. Aside from genetic lineage, the patient also underwent standard clinical ophthalmic examinations, including best-corrected visual acuity, slit-lamp biomicroscopy, dilated indirect ophthalmoscopy, fundus photography, fundus autofluorescence, and optical coherence tomography (OCT). The molecular diagnosis of BCD was performed as previously described[59] and the variants were confirmed by Sanger sequencing.

### Construction of plasmids
The sgRNAs were designed using Benchling (https://benchling.com) in intron 6 upstream the *CYP4V2* mutation c.802-8_810del17insGC. Seven complementary sgRNAs with 20 nucleotides adjacent to the 5'-end of the PAM site (NNGRRT) were generated with high specificity and efficiency (Supplementary Table 4). Each pair of sgRNA oligos were annealed and ligated into the pX601 plasmid (#61591; Addgene) between the Bsa1 restriction site, and the constructed vector was pX601-CMV-SaCas9-puro-sgRNA1-7.

Two types of donor plasmids were constructed. pMD™19-T-donor3/4-EGFP plasmid was used for HEK293T cells transfection and pX601-CMV-Puro-P2A-mCherry-donor3 was used for transfection of BCD-iPSCs. For the construction of pMD™19-T-donor3/4-EGFP plasmid, the inserted DNA fragment contained intron 6 from the matched sgRNA and the CDS of exon 7-11 of the *CYP4V2* gene, followed by EGFP. All three parts were sandwiched between sgRNA3/4 target sequences and then subcloned into Linearized pMD™19-T Vector (TAKARA 6013). For the construction of pX601-CMV-Puro-P2A-mCherry-donor3 plasmid, the inserted DNA fragment as described above was ligated to plasmid pX601 (#61591; Addgene). Then puro-P2A-mCherry cassette was inserted into the 5' end of the donor fragment, the mCherry does not insert into the repaired DNA sequence, it was used to mark the positive-transfected cells through fluorescent labeling.

To construct the AAV vector for in vivo study, the CMV promotor was replaced by the EFS promotor, and the puromycin fragment was removed, the constructed vector was pX601-EFS-SaCas9-sgRNA3. The EFS promotor was verified to express at RPE cells and the OS/IS and ONL layers which is consistent with other study[60]. The vector without sgRNA3 was also constructed as a control, i.e., pX601-EFS-SaCas9-blank. For the corresponding donor plasmid, the CMV promotor was also replaced by the EFS promotor, the constructed vector is pX601-EFS-Puro-P2A-mCherry-donor3. Construct the pAV-CAG-CYP4V3(E1-6)-CYP4V2(E7-11) plasmid containing desired transcription sequence based on pAV-CAG-CYP4V2 from previous study[20]. The fragment of CYP4V2(E1-6) was replaced by CYP4V3(E1-6) (NM_133969.3).

### Screening and assessment of sgRNA and donor
To compare the cutting efficiency of different sgRNAs, HEK293T cells in a six-well dish were transfected with 2.0 μg plasmid DNA of pX601-CMV-SaCas9-puro-sgRNA1-7 or control vector, together with 1 μg/μl polyethylenimine (PEI) transfection reagent (B600070, ProteinTech Group, Chicago, IL, USA) according to the manufacturer's protocol, and the ratio of DNA(μg) to PEI(μg) is 1:3. The medium with 2.5 mg/ml puromycin (A1113803, Life, USA-15 days) were refreshed daily from

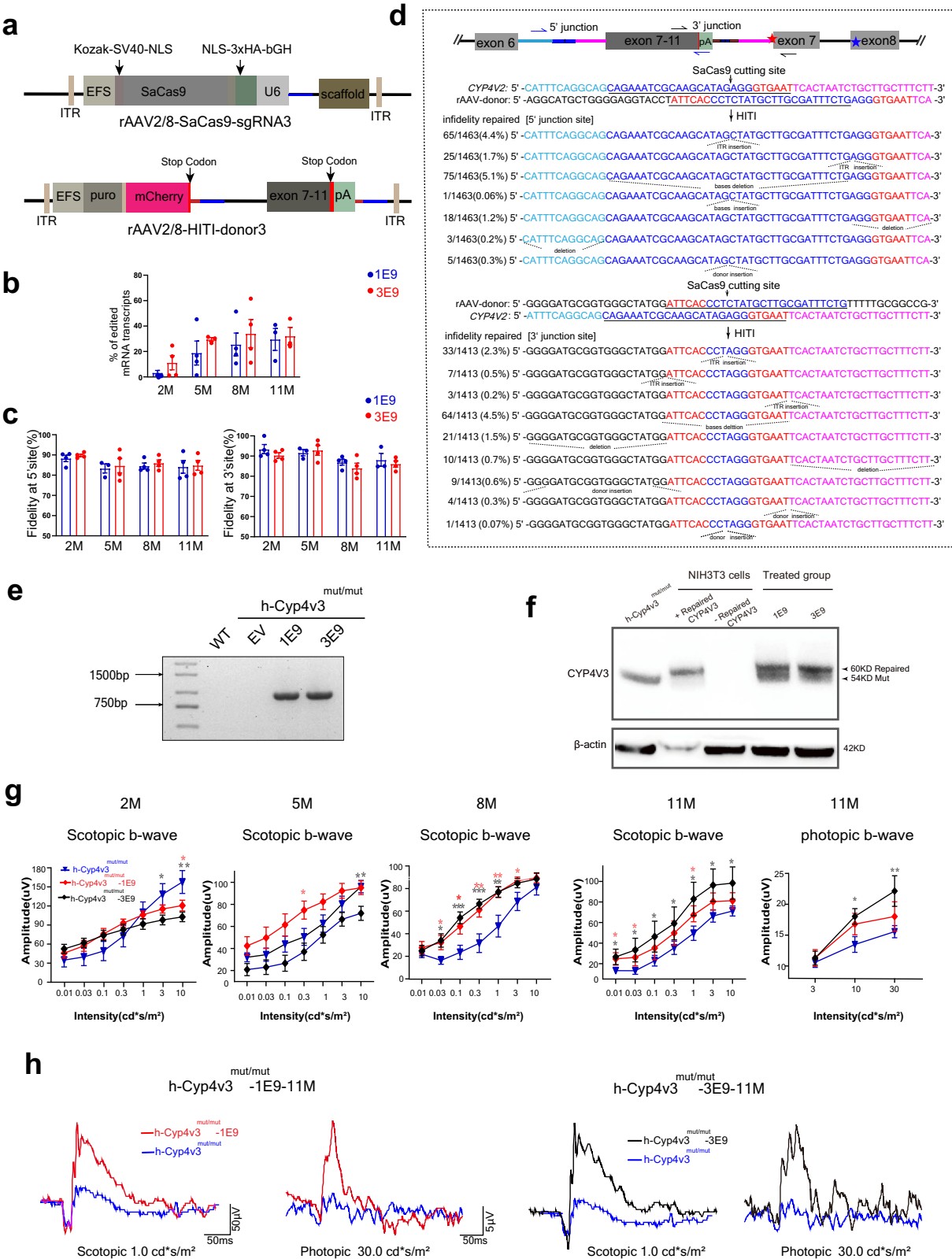

Day 3 to Day 7 post-transfection. On Day 7, the resistant cells were harvested. The cellular genomic DNA was extracted using the FastPure Cell/Tissue DNA Isolation Mini Kit (DC102-01, Vazyme, Nanjing, China), subjected to T7E1 assay (M0302S, NEB, USA) following the manufacturer's instructions, and captured on 2.0% agarose gels using a Gel Doc imaging system (Bio-Rad, USA). All primers used for PCR amplification are listed in Supplementary Table 1.

To test the targeted integration efficiency in vitro, HEK293T cells were transfected and puromycin (A1113803, Life, USA-15 days) selected positive cells for DNA extraction according to the manufacturer's protocol. For each well in a six-well plate, 2 μg pX601-CMV-SaCas9-puro-sgRNA3/4 and 2 μg pMD™19-T-donor3/4-EGFP were transfected together. To test the proportion of fidelity editing in vivo, DNA was extracted from retinal-RPE complexes of posttreatment mouse in 2, 5,

**Fig. 5 | HITI-based method achieved precise genome editing and restored visual function in *h-Cyp4v3^mut/mut* mice. a** Schematic diagram of rAAV2/8-SaCas9-sgRNA3 and rAAV2/8-HITI-donor3. **b** Proportion of edited mRNA transcripts at 2, 5, 8, and 11 months after treatment (i.e., 3, 6, 9, 12 months after birth) in *h-Cyp4v3^mut/mut*-1E9 (*n* = 4,4,4,3 eyes) and *h-Cyp4v3^mut/mut*-3E9 (*n* = 4,3,4,3 eyes) mice. Error bars represent the mean ± SEM. **c** The fidelity repair rate at 5′ junction sites of *h-Cyp4v3^mut/mut*-1E9/3E9 (*n* = 4,3,4,4; *n* = 4,4,4,4 eyes) mice, and 3′ junction sites of *h-Cyp4v3^mut/mut*-1E9/3E9 (*n* = 4,3,4,3; *n* = 4,4,4,4 eyes) mice at 2, 5, 8, 11 months after treatment. Error bars represent the mean ± SEM. **d** Schematic and sequencing analysis of 5′ and 3′ ends of the infidelity integration of *CYP4V2* after treatment in *h-Cyp4v3^mut/mut*. The arrows indicate the primers used to amplify the junction area flanked on endogenous *CYP4V2* and the HITI donor after integration. **e** Repaired mRNA sequence can be amplified by primers flanked on mE2 and hE9 in the treatment group (1E9 and 3E9) but not detected in the WT and empty vector group. EV, empty vector group. **f** Western blot results showed that two blots were shown in the treated group, the shorter one is in line with the truncated protein from humanized mouse, and the larger one is in line with the blot from NIH3T3 cells transfected with plasmid pAV-CAG-CYP4V3(E1-6)-CYP4V2(E7-11) containing the desired edited sequence. **g** Statistic graph of ERG amplitude in *h-Cyp4v3^mut/mut*-3E9 mice (*n* = 17, 20, 14, 7 eyes), *h-Cyp4v3^mut/mut*-1E9 mice (*n* = 26, 20, 14, 9 eyes) at 2, 5, 8, 11 months after treatment and age-matched *h-*

*Cyp4v3^mut/mut* mice (*n* = 17, 34, 14, 11 eyes). Student's unpaired two-tailed *t*-test. Error bars represent the mean ± SEM. Exact *P*-values: *h-Cyp4v3^mut/mut* mice vs. *h-Cyp4v3^mut/mut*-1E9 mice (10.0 cd*s/^m2 scotopic b-wave at 2 months after treatment) *p* = 0.0441; (0.30 cd*s/^m2 scotopic b-wave at 5 months after treatment) *p* = 0.0454; (0.03 cd*s/^m2, 0.10 cd*s/^m2, 0.30 cd*s/^m2, 1.0 cd*s/^m2 and 3.0 cd*s/^m2 scotopic b-wave at 8 months after treatment) *p* = 0.037, *p* = 0.0195, *p* = 0.00855, *p* = 0.00725, *p* = 0.0438; (0.01 cd*s/^m2, 0.03 cd*s/^m2 and 1.0 cd*s/^m2 scotopic b-wave at 11 months after treatment) *p* = 0.0168, *p* = 0.0495, *p* = 0.0465. *h-Cyp4v3^mut/mut* mice vs. *h-Cyp4v3^mut/mut*-3E9 mice (3.0 cd*s/^m2 and 10.0 cd*s/^m2 scotopic b-wave at 2 months after treatment) *p* = 0.0371, *p* = 0.00835; (10.0 cd*s/^m2 scotopic b-wave at 5 months after treatment) *p* = 0.00799; (0.03 cd*s/^m2, 0.10 cd*s/^m2, 0.30 cd*s/^m2 and 1.0 cd*s/^m2 b-wave at 8 months after treatment) *p* = 0.0158, *p* = 0.000827, *p* = 0.000889, *p* = 0.00668; (0.01 cd*s/^m2, 0.03 cd*s/^m2, 0.10 cd*s/^m2, 0.30 cd*s/^m2, 1.0 cd*s/^m2, 3.0 cd*s/^m2 and 10.0 cd*s/^m2 scotopic b-wave at 11 months after treatment) *p* = 0.0297, *p* = 0.0293, *p* = 0.0480, *p* = 0.0413, *p* = 0.0247, *p* = 0.0326, *p* = 0.0472; (10.0 cd*s/^m2 and 30.0 cd*s/^m2 photopic b-wave at 11 months after treatment) *p* = 0.0312; *p* = 0.00605.*p* < 0.05, **p* < 0.01, ***p* < 0.001. **h** Representative scotopic (intensity of 1.0 cd*s/m²) and photopic (intensity of 30.0 cd*s/m²) ERG waveforms of *h-Cyp4v3^mut/mut*-1E9 and *h-Cyp4v3^mut/mut*-3E9 mice at 11 months after treatment. Source data are provided as a Source Data file.

8, and 11 months using the FastPure Cell/Tissue DNA Isolation Mini Kit (DC102-01, Vazyme). In order to analyze the sequences carrying insertion, PCR amplification of the 5′ and 3′ junction area (Supplementary Table 1) ranging from genomic region to inserted sequence were subcloned into pMD™19-T Vector (TAKARA 6013). Molecular cloning and single colony sequencing were operated according to the manufacturer's protocol. In order to analyze those sequences without successful insertion in vivo, DNA extracted from retinal-RPE complexes of posttreatment 11 months mouse were amplified flanking target site by nested PCR (Supplementary Table 1). PCR amplicons were analyzed by NGS using the Illumina NovaSeq platform (Illumina, San Diego, CA, USA). The targeted integration was further verified at the protein level by Western blot.

To analyze whether the edited DNA sequences yielded aberrant RNA splice sites, four minigene plasmids were constructed. pMD™19-T-mgCYP4V2_exon6-intron6-exon7 (pmg_WT) was generated as the positive control, with *CYP4V2* DNA fragment (exon 6-intron6-exon7, NG_007965.1) inserted into pMD™19-T Vector (TAKARA 6013). pMD™19-T-mgCYP4V2_exon6-c.802-8_810del17insGC-exon7 (pmg_mut) was generated as the negative control, with *CYP4V2* DNA insertion fragment ranged from exon 6 to exon 7 with mutation of c.802-8_810del17insGC. pMD™19-T-mgCYP4V2_exon6-sgRNA3 stub duplication-exon7 (pmg_sgRNA3) mimics DNA sequence after site-directed cleavage and repair with sgRNA3, and pMD™19-T-mgCYP4V2_exon6-sgRNA4 stub duplication-exon7 (pmg_sgRNA4) mimics DNA sequence after site-directed cleavage and repair with sgRNA4, both DNA fragments ranged from exon 6 to exon 7 with corresponding stub duplication inserted into pMD™19-T Vector (TAKARA 6013). Four plasmids were transfected and puromycin (A1113803, Life, USA-15 days) selected in HEK293T cells with 2.0 μg DNA plasmids. RNA was isolated from transfected cells for further RT-PCR and sequencing analysis as below (Supplementary Table 1).

### iPSCs reprogramming and cell transfection

Urine cells (UCs) from the BCD patient and healthy control mentioned above were cultured and reprogrammed into iPSCs by retroviral transduction of Oct3/4, Sox2, Klf4, and c-Myc using the ReproEasy iPSC Reprogramming Kit (CA5002002, Cellapy Biotechnology, Beijing, China)[61–63]. The morphological features and markers were observed using a Nikon fluorescence microscope (Nikon, Japan). BCD-iPSCs were transfected with pX601-CMV-SaCas9-puro-sgRNA3 and pX601-CMV-Puro-P2A-mCherry-donor3 with a ratio of 1:1, a total of 5 μg plasmids using Lipofectamine Stem Reagent (STEM00003, Thermo Fisher Scientific, MA, USA) according to the manufacturer's instructions. The medium with 2.5 mg/ml puromycin (A1113803,

Life, USA-15 days) were refreshed daily from Day 3 to Day 7 post-transfection. Then, the puromycin was removed and the resistant single-cell was sorted and seeded into 96-well plates to generate single-cell clones. After culturing for 7d, cells of each well were harvested partly for genomic DNA extraction using the FastPure Cell/Tissue DNA Isolation Mini Kit (DC102-01, Vazyme). Two pairs of PCR primers (Supplementary Table 1) were designed for the edited/unedited genomic region amplification and analysis using a Gel Doc imaging system (Bio-Rad, USA). The above seeding process for positive-transfected iPSC cells was repeated until the single-cell clones appeared.

NIH3T3 cells were transfected with 2.0 μg plasmid DNA of pAV-CAG-CYP4V3(E1-6)-CYP4V2(E7-11), together with 1 μg/μl poly-ethylenimine (PEI) transfection reagent (B600070, ProteinTech Group) according to the manufacturer's protocol, and the ratio of DNA(μg) to PEI(μg) is 1:3. The control group are NIH3T3 cells without transfection. The medium was refreshed 6 h post-transfection and cells were harvested at 48 h for Western blot described below.

### RPE differentiation and cell apoptosis staining

The retinal organoids were generated referring to the articles[64,65]. The RPE cells were detached from the RPE domains of the NR-RPE organoids with good shapes using a tungsten needle. The isolated RPE cells were then seeded in 96-well plates at a density of 5000 cells in 100ul per well. Each iPSC-RPE cells were divided into 6 groups and each group was repeated three times, the medium is refreshed every other day. Then 50 μL medium was removed from RPE cells at different times of 2 h, 1 d, 2 d, 3 d, 4 d, and 5 d in different groups, and 10 μL CCK8 (Dojindo Laboratories, Japan) and 40 μL new medium were added to 100 μL according to the manufacturer's instructions. The OD values at OD450nm were detected with a microplate reader (BioTek Synergy H1, USA) for each group.

### Animals

All mice were bred and maintained at the Peking University Health Science Center Animal Care Services Facility in Specific Pathogen Free (SPF) conditions under a 12-h light/12-h dark cycle with ad libitum access to food and water. All animals were maintained in accordance with the guidelines of the Association for the Assessment and Accreditation of Laboratory Animal Care. All experiments were performed in accordance with the Association for Research in Vision & Ophthalmology (ARVO) Statement for the Use of Animals in Ophthalmic and Vision Research. Mice were randomly distributed to experimental groups, with half males and half females.

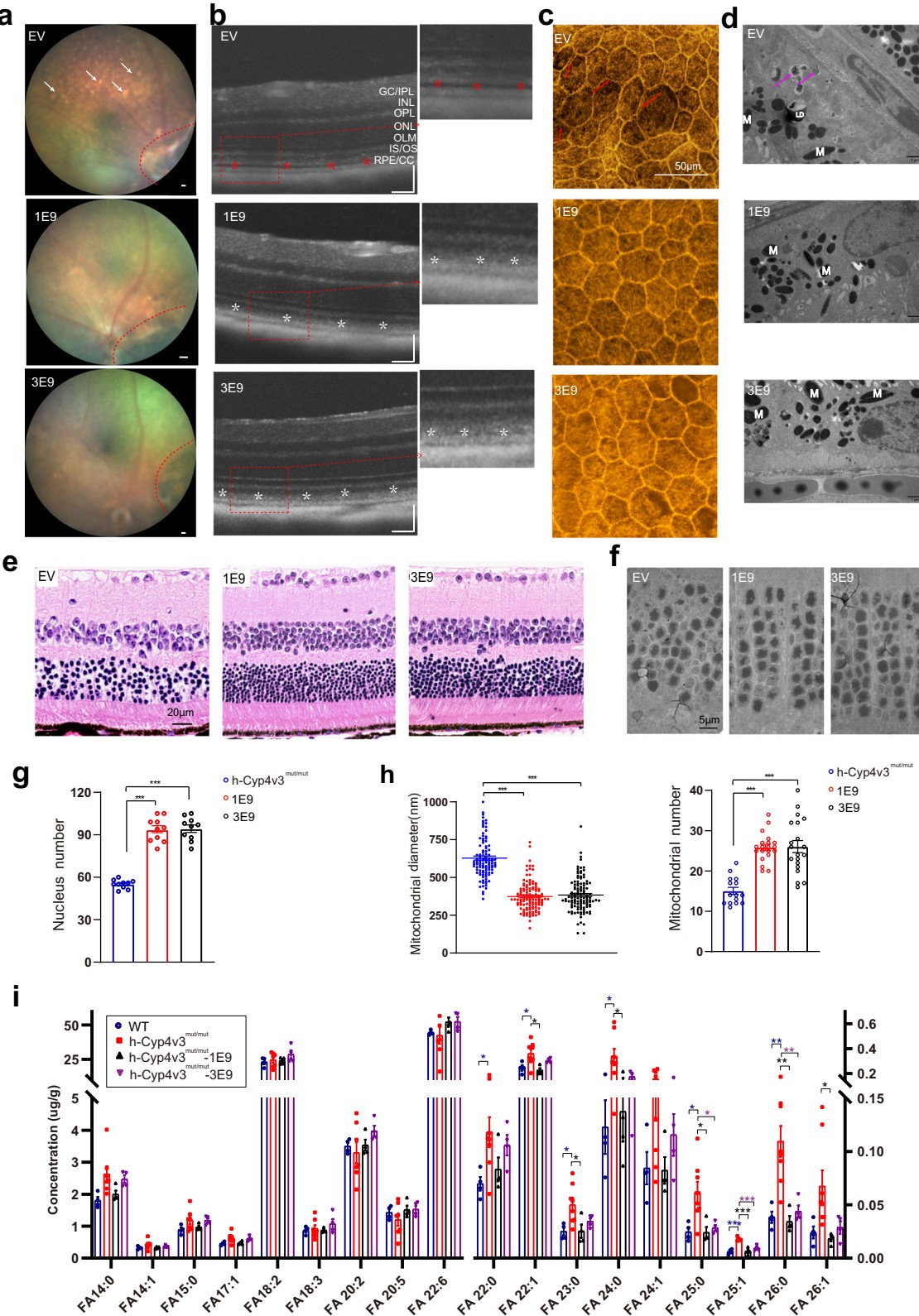

The *h-Cyp4v3^{mut/mut}* mice used in this study were custom-designed and obtained from Beijing Biocytogen Co., Ltd (Beijing, China). These models were generated by CRISPR/Cas9 technology based on homology-directed repair (HDR). In this mouse model, the m*Cyp4v3* genomic DNA from exon 6 (mE6) to exon 8 (mE8) was replaced with the human counterpart that contained two common clinical mutations: E7: c.802-8_810del17insGC and E8: c.992A>C simultaneously. A

pair of sgRNAs targeting exon 6-8 of the *Cyp4v3* gene and a homologous recombination template were designed. The sequence of sgRNAs is as follows, sgRNA for exon 6: 5′-AGAAGGACGGGACCAC AAAA-3′; sgRNA for exon 8: 5′-CTTCTGGATTCGTGCCCAAT-3′. The mRNA of in vitro transcribed Cas9 and sgRNAs along with homologous recombination template were injected into zygotes of C57BL/6J mice. After injection, the zygotes were implanted into the oviduct of

**Fig. 6 | HITI-based genome editing rescued RPE and photoreceptor degeneration in *h-Cyp4v3^{mut/mut}* mice at 11 months after treatment with rAAV2/8-SaCas9-sgRNA3 and rAAV2/8-HITI-donor3. a** Representative fundus images, scattered yellow-white granular deposits were found in EV group, the phenomena was rescued after 1E9/3E9 dose treatment. Red dotted lines indicated the injection area. EV empty vector group. Scale bars: 50 μm. **b** OCT images showed hypo-reflective lesions between the EZ and IZ in the EV group, and this lesion was not found after treatment. GC/IPL ganglion cell/inner plexiform layer, INL Inner Nuclear Layer, OPL Outer Plexiform Layer, ONL Outer Nuclear Layer, OLM Outer Limiting Membrane, IS/OS Inner segment/Outer segment, RPE/CC Retinal Pigment Epithelium/Choriocapillaris. Scale bars: 50 μm. **c** Phalloidin staining in RPE flat mounts showed 'railroad tracks'-like arrangement of adjacent RPE cells, which was alleviated after treatment. **d** TEM images in the RPE region showed accumulation of lipid droplets (LDs) in RPE cells and excessive metabolic residue deposition on the basal membrane, indicated by purple arrowheads, these lesions were not found after treatment. M melanin pigment granule. **e, f** H&E and TEM images of ONL showed that the number of nuclei decreased and the morphology was disrupted in the EV group, and the phenomena alleviated after treatment. Scale bars: 20 μm (**e**). Scale bars: 5 μm (**f**). **g, h** Statistical diagram of the number of nuclei in ONL per field (50 μm × 50 μm), 250 μm away from the ONH in retinal sections (*n* = 10 eye sections) (**g**), and quantification of the mitochondrial diameter and numbers in photoreceptor cells (**h**). *N* = 108,122 and 116 mitochondria were analyzed in *h-Cyp4v3^{mut/mut}*, 1E9, and 3E9 groups for mitochondrial diameter analysis; twenty (1E9,3E9) and fifteen (*h-Cyp4v3^{mut/mut}*) slices were used for analyzing the mitochondrial numbers. **i** Total FFA profile in the RPE-choroid complexes of *h-Cyp4v3^{mut/mut}* mice at 11 months after treatment and age-matched WT. WT (*n* = 4 eyes), *h-Cyp4v3^{mut/mut}* (*n* = 8 eyes), *h-Cyp4v3^{mut/mut}*-1E9 (*n* = 4 eyes), *h-Cyp4v3^{mut/mut}*-3E9 (*n* = 4 eyes) mice. 1E9 represent *h-Cyp4v3^{mut/mut}*-1E9 mice, 3E9 represent *h-Cyp4v3^{mut/mut}*-3E9 mice. Student's unpaired two-tailed *t*-test. Error bars represent the mean ± SEM. Exact *P*-values: **g** *h-Cyp4v3^{mut/mut}* mice vs. *h-Cyp4v3^{mut/mut}*-1E9 mice *p* = 9.384e-11, *h-Cyp4v3^{mut/mut}* mice vs. *h-Cyp4v3^{mut/mut}*-3E9 mice *p* = 3.76e-11. **h** mitochondrial diameter, *h-Cyp4v3^{mut/mut}* mice vs. *h-Cyp4v3^{mut/mut}*-1E9 mice *p* = 1.491e-40, *h-Cyp4v3^{mut/mut}* mice vs. *h-Cyp4v3^{mut/mut}*-3E9 mice, *p* = 1.167e-34; mitochondrial numbers, *h-Cyp4v3^{mut/mut}* mice vs. *h-Cyp4v3^{mut/mut}*-1E9 mice *p* = 1.544e-10, *h-Cyp4v3^{mut/mut}* mice vs. *h-Cyp4v3^{mut/mut}*-3E9 mice *p* = 1.675e-6. **i** *P*-values are listed in source data. *$p < 0.05$, **$p < 0.01$, ***$p < 0.001$. Source data are provided as a Source Data file.

pseudopregnant mice, and founder mice were obtained by natural birth. The obtained founder mice were validated by PCR, RT-PCR, Western blot for the targeted genome, transcription, and translation (Fig. 3b, c, Supplementary Table 1). The model was also excluded from the retinal degeneration mutations (*Pde6b^{rd1}*, *Crb1^{rd8}*, *Pde6b^{rd10}*, and *Rpe65^{rd12}*) using sanger sequencing (Supplementary Table 1) and potential off-target sites through Cas-OFFinder (Supplementary Table 2). The WT mice used in this study were C57BL/6J, which were purchased from the Beijing Vital River Laboratory Animal Center (Beijing, China).

### rAAV2/8 production
Recombinant AAV2/8 (rAAV2/8) was packaged and purified by Beijing Chinagene Corporation Ltd (Beijing, China). Briefly, human HEK293T cells were seeded in 15 cm plates with DMEM+10% fetal bovine serum (FBS)+1% penicillin/streptomycin with a density of 100,000 cells/plates initially, incubated at 37 °C in 5% $CO_2$ for three days until the cells covered 80% area of plates. The transgene plasmids pX601-EFS-SaCas9-sgRNA3, pX601-EFS-SaCas9-blank, and pX601-EFS-Puro-P2A-mCherry-donor3 were used as the viral genome to produce rAAV2/8-SaCas9-sgRNA3, rAAV2/8- SaCas9-blank and rAAV2/8-HITI-donor3 respectively. The transfection medium is prepared by mixing transgenic plasmid with packaging plasmids of pHelper and pAAV-RC8[30] at a ratio of 1:1:1 to 2 g and incubated at 37 °C in 5% $CO_2$ for 8 h. The transfection medium was changed to culture medium and incubated for 64 h at 37 °C in 5% $CO_2$. Transfected HEK293T cells were then lysed and clarified, followed by purification using affinity chromatography, ion exchange chromatography, ultrafiltration, and sterile filtration. The formulation buffer consisted of 10 mM PB buffer, 150 mM sodium chloride, 0.001% poloxamer 188, pH 7.3. Genomic titers were determined by qPCR.

### Subretinal injection
Mice were randomly allocated to three groups (low dose at $1 \times 10^9$ vg/eye, high dose at $3 \times 10^9$ vg/eye, and empty vector group at $1 \times 10^9$ vg/eye), and subretinal injection was performed on postnatal day 30–40. In accordance with the guidelines of 3R principle (Reduction, Replacement, Refinement) for laboratory animals, mice with bilateral eyes were included in the experimental group. Each eye (*n* = 44) received 2 μL of co-delivered AAV vectors mixed 1:1 at different concentrations. For treatment group, rAAV2/8-SaCas9-sgRNA3 and rAAV2/8-HITI-donor3 were used, for the empty vector group, rAAV2/8-SaCas9-blank, and rAAV2/8-HITI-donor3 were used. In the low-dose treatment group, the concentrations of the two viral vectors were $1 \times 10^9$ vg/μL respectively, with a total load of $2 \times 10^9$ vg. In the high-dose treatment group, the concentrations of the two viral vectors were $3 \times 10^9$ vg/μL respectively, with a total load of $6 \times 10^9$ vg. In the empty vector group, the concentrations of the two viral vectors were $1 \times 10^9$ vg/μL respectively, with a total load of $2 \times 10^9$ vg. Two animals with vitreous hemorrhage caused by the operation were excluded, and sixty-four animals with successful injection were included for further evaluation. The pupils of each mouse were dilated with 1% topical atropine (Alcon Laboratories, Fort Worth, TX, USA) and then anesthetized with an intraperitoneal injection of avertin (T48402, Sigma Aldrich, USA) dissolved in tertiary amyl alcohol. Under an ophthalmic microscope (Topcon, Japan), a small incision was made through the cornea adjacent to the limbus using a 30-gauge needle. A 33-gauge blunt needle (Hamilton, Switzerland) fitted to a Hamilton syringe was inserted through the incision while avoiding the lens and injected into the subretinal space. All the substances injected were administered with 0.1% fluorescein (100 mg/ml AKFLUOR, Alcon, Fort Worth, TX, USA) in order to visualize the surgical procedure. Following all injections, 1% atropine eye drops and neomycin-polymyxin B-dexamethasone ophthalmic ointments were applied.

### Fundus photography, optical coherence tomography, and electroretinogram (ERG)
Fundus photography and optical coherence tomography (OCT) examinations were performed using a Micron IV retinal imaging system (Phoenix-Micron, NW York Drive, USA). The pupils were dilated with 1% topical atropine for at least 10 min. The mouse was then anesthetized with an intraperitoneal injection of avertin dissolved in tertiary amyl alcohol. Methylcellulose solution (1%) was applied on the corneal surface to keep the eyes moist. The mouse was placed on the stage of the imaging system and images of the mouse retina were visible in the bright-field image control area. One hundred OCT images were accumulated to enhance the quality of the resulting images. The entire retina including the peripheral area could be observed by changing the angle between the camera and the eye. Fundus photography and OCT images were viewed and photographed using Micro IV software.

According to the International Society for Clinical Electrophysiology of Vision (ISCEV) standard[66], ERGs were recorded using the Espion E2 recording system (Diagnosys LLC, Lowell, MA, USA) at 3, 6, 9 and 12 months after birth, i.e. 2, 5, 8 and 11 months after treatment. All mice were dark-adapted overnight, and all the procedures were performed under a dim red light. After dilating the pupils with 1% topical atropine, the mouse was anesthetized with avertin by intraperitoneal injection and was placed on the platform at 37 °C with electrodes over the corneas. Scotopic recordings were performed at incremental light intensities of 0.01, 0.03, 0.10, 0.30, 1.0, 3.0, and 10.0 cd.s/m² for the 7-step protocol. Photopic recordings were performed at light

intensities of 3, 10, 30 cd.s/m$^2$ for three steps. Two additional flash intensities of 3 and 10 cd.s/m$^2$ were recorded for two steps. The amplitudes in the scotopic and photopic ERG responses were statistically analyzed.

## Histologic analysis and Western blot

For immunofluorescence of cryosections, the eyeballs were extracted from sacrificed mice and fixed in 4% paraformaldehyde for 1–2 h. Following the removal of the anterior segments and lens, the eye cups were left in the 30% sucrose for dehydration for 1 h. The eye cups were embedded into the optical cutting temperature medium (OCT, 4583, Tissue-Tek; Sakura Finetek, Torrance, CA) for flash frozen, and 7-μm cryosections around the optic nerve were cut in the sagittal orientation using a cryostat (Leica, Germany). Then slides were washed in PBS blocked in 5% donkey serum in PBS containing 0.1% Triton X-100 (PBST) for 1 h, followed by incubation with primary antibodies diluted in 5% donkey serum at 4 °C overnight. After washing, the slices were incubated with secondary antibodies and 0.2 μg/ml DAPI (1:5000; C0060, Solarbio,656 Beijing, China) for 1 h at room temperature. Images were captured using a confocal scanning microscope (A + /AIR+, Nikon, Japan)/an inverted microscope (Axio Vert.A1, Zeiss, Germany). The immunofluorescence staining of cells share similar staining and figure collection steps as cryosections.

For histological analyses, fixed eyes were dehydrated, embedded in paraffin, cut around the optic nerve using Leica slicing machine (Leica, Germany), and stained with hematoxylin and eosin (H&E). Images were photographed and measured using NanoZoomer Digital Pathology (Hamamatsu Photonics, Hamamatsu City, Japan).

For RPE flat mounts staining, the RPE flat mounts were dissected from the eyeballs and fixed in 4% paraformaldehyde for 1–2 h. After washing with PBS, the RPE flat mounts were incubated with TRITC-phalloidin (1:200; CA1610, Solarbio, Beijing, China), mounted, and observed using a microscope (Nikon, Tokyo, Japan).

For Western blot, cells or mouse retinas extracted from sacrificed mouse following removal of the anterior segments and lens were homogenized in radio immunoprecipitation assay (RIPA) buffer containing a proteinase inhibitor cocktail in a homogenizer, following the same steps as described[20]. The targeted bands were detected using enhanced chemiluminescence (ECL; Millipore, MA, USA) exposure reagent and collected using a ChemiDoc™ MP imaging system (Tanon Science & Technology Co., Shanghai, China).

## Transmission electron microscopy (TEM) and semi-thin section.

For light microscopy and transmission electron microscopy, enucleated eyes were fixed in 1% formaldehyde, 2.5% glutaraldehyde in 0.1 M cacodylate buffer (pH 7.5) for 1 h. Then, the cornea, iris, and lens were removed, and the eye cups were incubated in the same fixative at 4 °C overnight. Remove the sclera, leave choroid-RPE-retinal tissue complexes, and cut into 3*3 mm blocks. Eye cups were washed with buffer, post-fixed in osmium tetroxide, dehydrated through a graded acetone series, and embedded in Epon. Semi-thin sections (500 nm) were cut for light microscopy observations. For TEM, ultrathin sections (70 nm) were stained in uranyl acetate and lead citrate before viewing on an electron microscope (JEM-1400PLUS, Japan). The mitochondrial diameters and numbers were measured and recorded using ImageJ.

## Real-Time PCR

RNA was extracted using TRIzol Reagent (15596018CN, Thermo Fisher), following the One-Step gDNA Removal and cDNA Synthesis Supermix Kit (AT311, TransGen Biotech, Beijing, China) at a final volume of 20 μL. The Real-time PCR reactions were performed following the manufacturer's protocol of TransStart Top Green qPCR SuperMix (AQ132, TransGen Biotech) and then running on the ABI7500 Real-Time PCR Detection System (Carlsbad, CA, USA).

## Antibodies

The primary antibodies used in this study were as follows: rabbit monoclonal anti-NANOG antibody (1:200; ab109250; Abcam, Cambridge, MA), rabbit polyclonal anti-OCT4 antibody (1:200; ab19857; Abcam), mouse monoclonal anti-SSEA4 antibody (1:200; sc-21704; Santa Cruz), mouse monoclonal anti-TRA-1-60 antibody (1:400; ab16288; Abcam), rabbit monoclonal anti-CRALBP antibody (1:100; A11649; ABclonal, China), rabbit monoclonal anti-PAX6 antibody (1:100; A7334; ABclonal), rabbit anti-CYP4V2 antibody (1:100; generated by AbMax Biotechnology Co., Ltd.), immunogen CYP4V2 (NP_997235.3,1a.a.-525a.a) full-length human protein. The antibody was verified in WB, ICC, IHC-P, and IHC-Fr, which were provided in the peer review file; rabbit monoclonal antibody against recombinant human β- actin (1:5000; AC026, ABclonal, Wuhan, China), mouse monoclonal antibody against EGFP (1:1000; ab184601, Abcam, Cambridge, United Kingdom); rabbit Anti-CRISPR-Cas9 antibody (1:5000; ab203943, Abcam, Cambridge, MA); Mouse Monoclonal antibody against α-fetoprotein (AFP) (1:100, MA5-14666, Thermo Fisher); Mouse Monoclonal antibody against α-smooth muscle actin (SMA)(1:200, 14-9760-82, Thermo Fisher); Rabbit Polyclonal antibody against beta Tubulin 3/ TUJ1 (1:200, PA5-85639, Thermo Fisher). The secondary antibodies were as follows: donkey anti-mouse IgG (H+L), Alexa Fluor 488 (1:800; A21202; Thermo Fisher, Waltham, MA, USA), donkey anti-rabbit IgG (H+L), Alexa Fluor 488 (1:800, A21206, Thermo Fisher), donkey anti-rabbit IgG (H+L), Alexa Fluor 568 (1:800; A10042; Thermo Fisher), donkey anti-mouse IgG (H+L), Alexa Fluor 568 (1:800; A10037, Thermo Fisher); donkey anti-rabbit IgG (H+L), Alexa Fluor 647 (1:800, A31573, Thermo Fisher); horse radish peroxidase (HRP)-conjugated Goat anti-mouse IgG antibody (1:5000, A0216, Beyotime, Shanghai, China) or Goat anti-rabbit IgG antibody (1:5000; A0208, Beyotime).

## Sample preparation, lipid extraction, and LC−MS/MS

For FFA analysis, RPE-choroid complexes from 12-month-old WT ($n = 4$), $h$-$Cyp4v3^{mut/mut}$ -EV ($n = 8$), $h$-$Cyp4v3^{mut/mut}$-1E9 ($n = 4$) and $h$-$Cyp4v3^{mut/mut}$-3E9 ($n = 4$) mice were detached into 1.5 ml centrifuge tubes and homogenized. Then 0.2 ml isopropanol and 0.2 ml acetonitrile were added into homogenized tissues, shaken for 1 h and centrifuged. Following this, 95 ul of the supernatant was added to 5 μL of internal standard (FFA19:0) for mass spectrometry analysis. The targeted peak area was calculated using Target Lynx quantitative software (Waters Corporation). The retention time allowed error was 15 s. Single point internal standard method was used to obtain quantitative results.

## GUIDE-Seq and in silico prediction of off-targets

In HEK293 cell lines, plasmids pX601-CMV-SaCas9-puro-sgRNA3 and pX601 (#61591; Addgene) (as blank control) were transfected respectively, and genomic DNA was isolated after transfection. The sequencing libraries were created as previously described[39] with minor modification: 400 ng of genomic DNA was sheared in 120 μl × 1 TE Buffer (Tris-EDTA: 10 mM Tris-base, 1 mM EDTA, pH 8.0) with adaptive focused acoustics (BioruptorTM Pico, Diagenode Belgium) to obtain 500 bp fragments. The sheared DNA was concentrated using AmPure XP beads (130 μl; Beckman Coulter) following the manufacturer's protocols, and eluted in 15 μl × 1 TE buffer. Then Nuclease-free H$_2$O (0.5 μl), dNTP mix (5 mM, 1.0 μl), Slow Ligation Buffer (10×, 2.5 μl), End-repair mix (low concentration, 2.0 μl), Buffer for Taq Polymerase (10×, Mg$^{2+}$ free, 2.0 μl) and Taq Polymerase (non-hot start, 0.5 μl) were added to the DNA sample (from first step, 14.0 μl) to a total of 22.5 μl, and the following reaction conditions were used: 12 °C, 15 min; 37 °C, 15 min; 72 °C, 15 min; 4 °C, Hold. Adapter ligation, targeted amplification, and GUIDE-Seq analysis were performed referring to the published article[39].

The Cas-OFFinder[38] online search tool (available at www.rgenome.net/cas-offifinder/) was used to search for potential SaCas9/SpCas9

off-target sites (PAM = 5′-NNGRRT-3′/ PAM = 5′-NGG-3′) in the human genome (GRCh38)/mouse genome (GRCm38), and the target-specific portion of the sgRNAs was used in this study as a query sequence, separately. Those highest-ranked potential off-target sites were chosen and examined (Supplementary Table 2,3). For sgRNA3, HEK293T cells were transfected by pX601-CMV-SaCas9-puro-sgRNA3 and the genomic DNA was isolated for PCR amplification and Next-generation sequencing. The next-generation sequencing method is referred to published articles[67–70].

**Statistics and Reproducibility.** All data were collected and analyzed using Excel 2019 and performed using Prism Software (GraphPad). The results were analyzed using the student's *t*-test. Data are presented as the mean ± standard error of the mean (SEM). Statistical significance was defined as $p < 0.05^*$, $p < 0.01^{**}$, $p < 0.001^{***}$.

Reproducibility of representative experiments: All phenotypes including fundus photography, ERG, and OCT examination were observed in at least three independent cohorts of animals. Images of immunoblots, gels, photomicrography, RPE flat mounts and immunofluorescence, HE staining, and TEM examination in histological sections represent at least two independent (biological) repetitions of experiments. The choice of all samples is unbiased. Sample sizes, statistical tests, and exact *P*-values are indicated in the figure legends and the source data file.

### Reporting summary

Further information on research design is available in the Nature Portfolio Reporting Summary linked to this article.

## Data availability

The human genome (GRCh38), mouse genome (GRCm38) data used in this study are available at https://www.ncbi.nlm.nih.gov/genbank/. The raw Sanger sequencing dataset needed to evaluate the conclusions in this study has been deposited in Figshare. The link is https://figshare.com/s/0e4c6fd93661c9944f63. All data generated in this study are provided in this manuscript and its Supplementary information/ Source Data file. Source data are provided with this paper.

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

## Acknowledgements

The authors thank all the patients who kindly participated in this study. This study was supported financially by National Natural Science Foundation of China (82371074, L.Y.), The 14[th] Five-Year Plan "Reproductive Health and Women and Children's health protection" key project (2023YFC2706304, L.Y.), National Key R&D Program of China (2023YFC3403302, S.C), Beijing Natural Science Foundation of Beijing-Tianjin-Hebei Basic Research Cooperation Project (J230031, L.Y.), Beijing Science and Technology plan project (Z231100004823021, L.Y.) and Peking University Third Hospital clinical key project (BYSYZD2022006, L.Y.).

## Author contributions

X.M. conceived and executed the project, analyzed the data, drafted and revised the paper; R.J., X.Z. supervised data analyses and drafted the

paper; F.Z., S.C., provided support with in vitro experiments; S.Y., X.L., H.D., X.F., and J.Z. provided support with experimental techniques; N.W., B.X. reviewed the manuscript; L.Y. designed the study and revised the manuscript. All authors have read and approved the article.

## Competing interests

F.Z., S.C., J.Z., and N.W. are current employees of Beijing Chinagene Co., LTD. X.Z is a former employee and was employed by Beijing Chinagene Co., LTD at the time this work was conducted. The authors have no other relevant affiliations or financial involvement with any organization or entity with a financial interest in or financial conflict with the subject matter or materials discussed in the manuscript apart from those disclosed. X.M., S.C., F.Z., and L.Y. have filed patents pertaining to the work described in this manuscript on behalf of Peking University Third Hospital and Beijing Chinagene Co., LTD. The patents have completed the transformation of relevant intellectual property rights from Peking University Third Hospital to Beijing Chinagene Co., LTD for future development. The remaining authors declare no other competing interests.
