## [Peer Review File · Nature Communications]

In vivo genome editing via CRISPR/Cas9 mediated homology-independent targeted integration for Bietti crystalline corneoretinal dystrophy treatmentREVIEWER COMMENTS

Reviewer #1 (Remarks to the Author):

This study offers an interesting approach to the treatment of CYP4V2-related BCD. The strategy is relevant and the data presented are encouraging. The creation of a novel mouse model enables in vivo outputs in addition to in vitro testing in HEKs and BCD patient-derived iPSCs and RPE organoids. Much of the data are of interest and indicate encouraging signs of success with the strategy but I feel there are gaps in the data set that need filling.

As a general query, why the use of AAV8 and not AAV2? Presumably the RPE is the primary target followed by photoreceptors, in which case AAV2 would be the expected serotype. What cells are being targeted? The donor vector carries an mCherry reporter and it would seem important to know what cells are being transduced and across what area of retina/RPE to achieve the outcomes presented. Use of the EFS promoter is intriguing as it is not known to provide a strong expression profile in the mouse retina so it would seem important to confirm the expression profile that is leading to the in vivo outcomes. It would be good to see mCherry and Cyp4v3 expression profiles by IHC. As the Cyp4v3 carries the C-terminus of human CYP4V2, presumably it may be possible to use a human-specific antibody?

Details of sample processing for western blot do not appear to be given (apologies if I have missed them), were the neural retinae lysed or the whole eye cup or the RPE only? Is any truncated protein produced from the donor transgene? For all western blots shown it would be good to have the ladder included and untrimmed versions placed in the supplementary materials.

cDNA analyses are provided following PCR and subcloning. However, whilst I appreciate the data from this, it does have its limitations in being selective and somewhat restrictive. More sequence details could be provided of the transcripts generated and other primer pairs could be used to better understand the breadth of transcripts being produced rather than selecting for those that carry the new insertion only.

The qPCR primer design is a bit unclear, the naming indicates the primers used to quantify edited vs unedited mRNA transcripts target either human CYP4V2 exon 10 (FW) and human CYP4V2 exon 11 (RV) or mouse exon 9 (FW) to mouse exon 10 (RV). If this is so, is it possible that the human targeting primer pair will amplify mRNA that is generated from uncut donor transgene? In this scenario, the calculation used to determine editing would be contaminated with donor transcripts.

We do not appear to see data indicating the on-target DNA editing success in vivo, which also seems a critical step and it would be of value to have off-target assessments from the in vivo study.

I wonder if some of the points above simply need clarifying but others need re-visiting as they would be critical in understanding the apparent positive outcomes observed with the degree of editing achieved and in what cell types.

Specific comments whilst working through the document:

- Title should not use "BCD" and should instead read "Bietti crystalline corneoretinal dystrophy".
- It could be made clear early on that Cyp4v3 is the murine ortholog of human CYP4V2.
- Line 46 "are suffered with" should be changed (e.g. exhibit, present or suffer).
- Line 49, the prevalence in Europe is given so the figure for East Asia should also be provided.
- Lines 70-72, reference for these data?
- Line 92 "we reported" should be "we report".
- Line 116, "high fidelity insertion" - this was achieved by PCR of the target region followed by subcloning for single colony sequencing analysis. This is fine for identifying the different insertion types (up to ~200bp?) but does not indicate rates of insertion.
- Figure 1c "n"?
- I'm not sure how relevant the mini gene assay is. If I am reading the details correctly, only the small changes at the insertion site are included in intron 6, which sort of makes sense but the reality will be that the coding sequence of exons 7-11 will be inserted. Would

transcript analysis on this would be more relevant? For example in the iPSC-derived RPE cells?

- Line 134, states the stub duplication does not affect normal mRNA splicing therefore all knock-in of the repaired sequences are considered precise. I don't think this extrapolation can be claimed.
- Line 136 refers to translation of the repaired CYP4V2 but this is not clear from the western blot. The blot is very trimmed and as a minimum the complete blot should be put in the supplementary. What size are we seeing? Where is the positive control (plasmid with CYP4V2 CDS)? Were both anti-CYP4V2 and anti-GFP used on this one blot? Why is there more CYP4V2 in HITI-4? Presumably repaired protein should also carry the GFP?
- Line 143 (and more), I think "bugle" is written instead of "bulge". This occurs a few times.
- Line 143, "No target that less than 3..." should be "No target with less than 3..."
- Line 167, were the RT-PCR products sequence confirmed?
- As previously, the western blot in figure 2 is very trimmed, could the original fully annotated version go in the supplementary?
- Were transcript and protein production in the RPE cells confirmed? Without confirmation of expression being maintained in these differentiated cells, the viability assay is of less value.
- Line 189, could sequencing evidence of the DNA and mRNA (following RT-PCR) please be included confirming the mutation and it's consequence.
- Figure 3e needs to be annotated so the layers of the retina are made apparent to those unfamiliar. I am not convinced the hyporeflexive lesions are very clear.
- Line 197, the lesions are described as "progressive and highly significant" but there is no quantification of this.
- Where are the data for the LC-MS/MS analysis?
- Same issue as before regarding the western blot in 5f.
- Line 270, I don't think there is evidence of normal transcription and translation. There is evidence of new mRNA transcripts and the western gives a hint of a larger protein. We do not know if this is the desired protein or how it compares to WT (at this point in the manuscript).
- Line 281, has it recovered? That seems surprising as postoperative loss of ERG tends to be maintained over time. Is it possible that the untreated cohort has reduced to the level of

functional reduction caused by an injection? Looking at the traces, I think if these were overlaid the empty vector injected cohort may be relatively stable from 2 months but the untreated lose function. How would it look if you plotted peak amplitudes at 1cdsm2 at each time point for each cohort?

- Line 297, why would the high dose group show a significant reduction in ERG at 5 months but this become a significant improvement from 8 months?

- Line 304, I'm sorry, but I struggle to see any features here that are clearly defined and different between untreated and treated.

- Line 351 onwards, I'm not sure it's relevant to mention the HMEJ work as this was not reported in the results.

Reviewer #2 (Remarks to the Author):

This is a very exciting research study by Meng et al. on the utility of HITI method of CRISPR/Cas9 in vivo gene editing. They have applied this technique to correct the most common genetic defect in the CYP4V2 gene-the causative agent for the degenerative eye disease Bietti's Crystalline Dystrophy (BCD). The efficacy of this system was demonstrated first using HEK cells followed by patient-derived iPSC cells differentiated to the appropriate ocular cell type-retinal pigmented epithelial cells. In vivo efficacy was finally demonstrated using a mouse model of BCD. There is great enthusiasm for this study and I only have minor comments which I will list below.

1. The authors state that "subretinal injection adversely affected on visual function..." What is the nature of this effect? Does retinal detachment occur? This needs to be clarified.

2. For the mouse study, 40 eyes per group were used. The Methods are unclear-did the authors use 20 mice per group or did they only inject one eye per mouse. The latter would be ideal in terms of safety data before initiating clinical trials. If the authors did not take this approach, justification should be provided.

3. The authors state that only animals with no surgical complications were included for further study-what were the number of animals and did it differ between groups?

4. The quality/specificity of the CYP4V2 antibody used in this study is concerning. The blot in figure 1G has quite a bit of non-specific binding. On the other hand, the blot in figure 3C is quite clean and a different banding pattern is seen in figure 5F. To my knowledge, none of

the mutations in the CYP4V2 gene produce detectable protein, including the H331P mutant <https://pubmed.ncbi.nlm.nih.gov/22772592/>. The authors need to address these discrepancies.

5. The results of the in vivo studies are very encouraging, including reconstitution of ocular responses and lipid metabolites. Thus, one assumes that the RPE have been transduced, edited and are producing functional CYP4V2 protein. But, to be definitive, the authors should provide immunohistochemical staining for CYP4V2 in ocular tissue sections.

Response to Reviewer Comments/Questions

Our response to each comment/query is shown in blue text, all changes in the manuscript (marked in red and italic) are also shown throughout the following response letter.

Reviewer #1: This study offers an interesting approach to the treatment of CYP4V2-related BCD. The strategy is relevant and the data presented are encouraging. The creation of a novel mouse model enables in vivo outputs in addition to in vitro testing in HEKs and BCD patient-derived iPSCs and RPE organoids. Much of the data are of interest and indicate encouraging signs of success with the strategy but I feel there are gaps in the data set that need filling.

Comment 1: As a general query, why the use of AAV8 and not AAV2? Presumably the RPE is the primary target followed by photoreceptors, in which case AAV2 would be the expected serotype. What cells are being targeted? The donor vector carries an mCherry reporter and it would seem important to know what cells are being transduced and across what area of retina/RPE to achieve the outcomes presented. Use of the EFS promoter is intriguing as it is not known to provide a strong expression profile in the mouse retina so it would seem important to confirm the expression profile that is leading to the in vivo outcomes. It would be good to see mCherry and Cyp4v3 expression profiles by IHC. As the Cyp4v3 carries the C-terminus of human CYP4V2, presumably it may be possible to use a human-specific antibody?

Response: Thanks a lot for your question. The targeted cells here are RPE and photoreceptors. We used AAV8 rather than AAV2 for the following reasons: Firstly, our previous study showed that serotype of AAV8 has higher transfection and expression efficiency to RPE and photoreceptor cells than AAV2 (Liu X et al., *eLife* 2023; Hu S et al., *Journal of Peking University Health sciences* 2020). Secondly, it is reported that neutralizing antibodies to AAV2 is the most common antibodies in all regions (Calcedo R., et al., *The Journal of infectious diseases* 2009). On the contrary, AAV8 was isolated from nonhuman primates, and antibodies to AAV8 is expected to have low serum prevalence among humans. There are increasing trends of using AAV8 rather than AAV2 in eye disease clinical trials such as Phase I/II dose escalation trial of A002 launched by MeiraGTx for Achromatopsia (NCT03001310); Phase I trial of RGX-314 launched by Regenxbio for Wet-AMD (NCT05407636); Phase I/II trial of CPK850 launched by Novartis for Retinitis Pigmentosa (NCT03374657); Phase I/II trial of ZVS101e launched by Chigenovo for Bietti crystalline corneoretinal dystrophy (NCT05714904) and Ongoing Safety and Tolerability Study of ZVS203e launched by Chigenovo for Retinitis Pigmentosa (NCT05805007).

Our group has explored EFS promoter expression profile in *RHO*-adRP gene editing therapy, the results showed that EFS promoter provide a strong and persistent expression profile in RPE and photoreceptor cells. This reference is added in the Methods on page 24 line 465 as follows: “*The expression profile of EFS promotor has been verified in our previous study⁶⁰.*”

We deeply appreciated your consideration of antibody. In fact, mouse CYP4V3 protein shares 92% similarity with human CYP4V2 protein, and there is no specific antibody to mouse CYP4V3. We failed to distinguish repaired CYP4V3 from endogenous CYP4V3 by human-specific antibody in IHC study as follows. Fortunately, the mutated and repaired protein has different molecular weight, which helps to make a distinction by Western blot in Fig.5f.

Comment 2: Details of sample processing for western blot do not appear to be given (apologies if I have missed them), were the neural retinae lysed or the whole eye cup or the RPE only? Is any truncated protein produced from the donor transgene? For all western blots shown it would be good to have the ladder included and untrimmed versions placed in the supplementary materials.

Response: We apologize for any confusion and sincerely appreciate your warm comments. The sample for western blot is the eyeballs extracted from sacrificed mouse following removal of the anterior segments and lens. It includes RPE layer accompanying with partly adhered choroid and neural retina. CYP4V3 is expressed mainly in RPE layer, and the stripping of its' adherent structures may damage it. The information was added on page33 line 649-651 as follows: *“For Western Blot, cells or mouse retinas extracted from sacrificed mouse following removal of the anterior segments and lens were homogenized in radio immunoprecipitation assay (RIPA) buffer containing a proteinase inhibitor cocktail in a homogenizer, following the same steps as described²⁰.”*

No truncated protein was produced from the empty vector group (injected with donor only) as following picture. It is consistent with the result (Comment 4) that there is no transcript generated from uncut donor transgene. The untrimmed western blot results were placed in the supplementary materials. Thank you for your kind suggestion.

Comment 3: cDNA analyses are provided following PCR and subcloning. However, whilst I appreciate the data from this, it does have its limitations in being selective and somewhat restrictive. More sequence details could be provided of the transcripts generated and other primer pairs could be used to better understand the breadth of transcripts being produced rather than selecting for those that carry the new insertion only.

Response: We greatly appreciated your profound comments. In order to minimize the following impact on transcription, we selected sgRNAs that do not affect mRNA splicing after cleavage in intron 6. To better understand various transcripts that do not achieved targeted editing, we designed three pairs of primers (exon 1-exon 9, exon 1-exon 10, and exon 1-exon 11) targeting mouse *Cyp4v3* cDNA and subcloning. Sanger sequencing detected no abnormal transcripts. We revised the manuscript on page 14 line 261-264 as follows: *“In addition to the successfully edited transcript,*

we designed three pairs of primers for mouse cDNA (Supplementary Table1) to better understand the unpredicted transcripts after cleavage. PCR and subclones sequencing (n=60) were performed on cDNA from the treated mouse eyes (n=9). The sequencing results showed no abnormal transcript production.”

Comment 4: The qPCR primer design is a bit unclear, the naming indicates the primers used to quantify edited vs unedited mRNA transcripts target either human CYP4V2 exon 10 (FW) and human CYP4V2 exon 11 (RV) or mouse exon 9 (FW) to mouse exon 10 (RV). If this is so, is it possible that the human targeting primer pair will amplify mRNA that is generated from uncut donor transgene? In this scenario, the calculation used to determine editing would be contaminated with donor transcripts.

Response: We sincerely appreciate the valuable comment. Firstly, I would like to explain the reason for designing primers. The transcripts difference in treated (mE1-6, hE7-11) and untreated humanized mouse (mE1-6, hE8, mE9-11) are the last three exons (E9-11). Quantification of both edited and unedited cDNA was measured using real-time qPCR with two primer pairs binding to hE10, hE11(edited sequence) and mE9, mE10(unedited sequence) respectively. The editing efficiency is calculated using the following formula: quantity of edited cDNA/ (quantity of edited cDNA+ quantity of unedited cDNA). **Your consideration of producing transcripts from donors is worth considering. To avoid the above issues, we conducted the same quantification and calculation on empty vector group (n=4) (injected with donor only), no more than 0.1% is shown, which can be negligible. We revised the manuscript accordingly to clarify above concern on page 12 line 233-235 as follows: “In order to quantify the potential transcript contamination caused by unedited donor sequences, we conducted the above calculation in the empty vector group (n=4, injected with donor only). The displayed value does not exceed 0.1% and can be ignored.”**

Comment 5: We do not appear to see data indicating the on-target DNA editing success *in vivo*, which also seems a critical step and it would be of value to have off-target assessments from the *in vivo* study. I wonder if some of the points above simply need clarifying but others need re-visiting as they would be critical in understanding the apparent positive outcomes observed with the degree of editing achieved and in what cell types.

Response: We apologize for any confusion. To test the on-target DNA editing success *in vivo*, firstly, we tried to design a pair of primers binding to mouse DNA sequence spanning the upstream and downstream of insertion site. In this scenario, there would be two different bands, one short unedited band (about 200bp) and one larger band including the insertion sequence (about 1700bp). However, gel electrophoresis did not show two different bands, mainly because the length difference between the fragments was too large, which led to the difference in amplification efficiency between the two PCR products. Secondly, we reference to Suzuki, K., et, al (*Nature*, 2016) and designed two paired primers on 5' and 3' junction sites separately. As shown in Fig.5d, there are two PCR products amplified from successfully edited DNA sequence. Only on-target successfully edited DNA sequence can be amplified and those two PCR products overlap the insertion sequence adequately. The following subcloning and sanger sequencing also revealed whether there is any unpredicted sequence at junctions caused by editing. But this method has its limitation. It just demonstrated the

exiting of successful insertion and the editing fidelity at the DNA level, we cannot calculate the exact editing efficiency. Thirdly, the editing efficiency is tested at cDNA level, please refer to response to Comment 4 for detail.

Comment 6: Specific comments whilst working through the document:

- Title should not use “BCD” and should instead read “Bietti crystalline corneoretinal dystrophy”.

Response: We sincerely thank you for careful reading and reminding. As suggested by the reviewer, we have corrected the title into “*In vivo genome editing via CRISPR/Cas9 mediated homology-independent targeted integration for Bietti crystalline corneoretinal dystrophy treatment*”.

Comment 7: - It could be made clear early on that Cyp4v3 is the murine ortholog of human CYP4V2.

Response: Thank you for your suggestion. We added this information in **Introduction on page 4 line 58-60** as follows: “*The Cyp4v3 gene is the mouse ortholog of human CYP4V2. The proteins share 82% identity and 92% similarity¹⁷ and the murine model can be used as an ideal disease model to implore BCD pathogenesis and therapy.*”

Comment 8: - Line 46 “are suffered with” should be changed (e.g. exhibit, present or suffer).

Response: Thank you for your reminder. We changed “are suffered with” to “*suffer*” as suggested accordingly.

Comment 9: - Line 49, the prevalence in Europe is given so the figure for East Asia should also be provided.

Response: Thank you. We added this information on **page 3 line 42-44** as follows: “*The estimated prevalence of BCD in Europe is 1/67000⁴. However, it is more common in East Asia, estimated to be 1/25000 in China, and is one of the most common pathogenic genes in Chinese inherited retinal dystrophies (IRDs) patients^{5,6}.*”

Comment 10: - Lines 70-72, reference for these data?

Response: Thanks for your suggestion. The published ARVO abstract “Safety and Efficacy of AAV-Mediated Gene Replacement Therapy in Bietti Crystalline Corneoretinal Dystrophy (BCD) Patients with CYP4V2 mutations” provides the reference. We added the related reference in the manuscript (**page 4 line 67**).

Comment 11: - Line 92 “we reported” should be “we report”.

Response: Thank you. This has been corrected.

Comment 12: - Line 116, “high fidelity insertion” - this was achieved by PCR of the target region followed by subcloning for single colony sequencing analysis. This is fine for identifying the different insertion types (up to ~200bp?) but does not indicate rates of insertion.

Response: Thank you for your suggestion. Indeed, the single colony sequencing analysis just be used to calculate the fidelity rate (proportion of expected repair in all repaired colonies) rather than insertion rate. The purpose of HEK293T cell experiment is to choose the suitable sgRNA and the corresponding donor for the following animal experiments. Considering the difference between *in vivo* and *in vitro*, combing the difference between plasmid transfection and virus infection, the editing efficiency was tested *in vivo* rather than *in vitro*.

Comment 13: - Figure 1c “n”?

Response: We apologize for the confusion. We repeated three times of T7E1 cutting and obtained the same trend result. We added this information n=3 on the Figure legend (**page 47 line 907**).

Comment 14: - I’m not sure how relevant the mini gene assay is. If I am reading the details correctly, only the small changes at the insertion site are included in intron 6, which sort of makes sense but the reality will be that the coding sequence of exons 7-11 will be inserted. Would transcript analysis on this would be more relevant? For example in the iPSC-derived RPE cells?

Response: Thank you. Your suggestion is really considerate and we also analyzed the mRNA sequence in edited iPSC using PCR (shown in Fig 2e) and Sanger sequencing (page 9 line 156 in manuscript and added in the supplementary materials), iPSC-derived RPE cells using Sanger sequencing (page 9 line 163 in manuscript and added in the supplementary materials) and in edited mouse using PCR and Sanger sequencing (page 13 line 258 in manuscript and added in the supplementary materials).

Comment 15: - Line 134, states the stub duplication does not affect normal mRNA splicing therefore all knock-in of the repaired sequences are considered precise. I don’t think this extrapolation can be claimed.

Response: Thank you for pointing this out. This statement has been deleted from the manuscript. More discussion about splicing is shown in Discussion on page 18 line 337-354 as follows: “*Precise genomic correction is the most important consideration for successful gene editing-based drug development. Indels at the target site may adversely affect the HITI-based gene therapy, such as mRNA splicing of the edited gene...*”

Comment 16: - Line 136 refers to translation of the repaired CYP4V2 but this is not clear from the western blot. The blot is very trimmed and as a minimum the complete blot should be put in the supplementary. What size are we seeing? Where is the positive control (plasmid with CYP4V2 CDS)? Were both anti-CYP4V2 and anti-GFP used on this one blot? Why is there more CYP4V2 in HITI-4? Presumably repaired protein should also carry the GFP?

Response: Thank you very much. The untrimmed western blot is placed in the supplementary materials with corresponding sizes (CYP4V2, 60kD; CYP4V2-GFP, 85kD). According to your suggestion, we added samples from HKE293T cells without any treatment as the positive control (which was shown as NC in Fig 1f), because HKE293T cells can express *CYP4V2* endogenously, there is no need to transfect CYP4V2 plasmid.

As for more CYP4V2 in HITI-4, there was a fingerprint in the HITI-4 group which cause confusion. We were really sorry for this mistake and replaced it with a clean background. The complete blot was shown in the supplementary.

Western blot presented in Fig 1g (anti-CYP4V2 rather than anti-GFP) can discriminate the endogenous CYP4V2 and a larger CYP4V2-GFP fusion protein. We apologize for the mistakes and revised it in the figure legend (page 47 line 915). The repaired protein carry GFP, which is help to screen better sgRNA from HITI-3 and HITI-4. According to your comment, we placed the blot using anti-GFP antibody in the Supplementary Fig.1b. The complete blot is also put in the supplementary.

Comment 17- Line 143 (and more), I think “bugle” is written instead of “bulge”. This occurs a few times.

- Line 143, “No target that less than 3...” should be “No target with less than 3...”.

Response: Thank you. All mistakes have been corrected accordingly.

Comment 18 - Line 167, were the RT-PCR products sequence confirmed?

Response: Thank you. Yes, the RT-PCR products was confirmed with sanger sequencing. We uploaded the sequences in the supplementary materials and added it in the manuscript.

Comment 19- As previously, the western blot in figure 2 is very trimmed, could the original fully annotated version go in the supplementary?

Response: Thank you. The original fully annotated version was uploaded in the supplementary materials.

Comment 20- Were transcript and protein production in the RPE cells confirmed? Without confirmation of expression being maintained in these differentiated cells, the viability assay is of less value.

Response: Thank you. These RPE cells were derived from edited iPSCs, the edited sequence was

confirmed with Sanger sequencing on DNA/mRNA level and the edited protein was confirmed with Western Blot in iPSCs, which is shown in Fig 2e-f and supplementary materials. Transcript in the RPE cells has been confirmed by sanger sequence. According to your suggestion, the information was added in Result on page 9 line 162 as follows: “..., *from which RPE cells were isolated, identified (Supplementary materials) and evaluated.*” Because of the shortage of edited iPSC-RPE protein sample and at least 6 months from iPSC editing to RPE differentiation, the protein experiment was not provided.

Comment 21- Line 189, could sequencing evidence of the DNA and mRNA (following RT-PCR) please be included confirming the mutation and it’s consequence.

Response: Thank you. These sequences were added in the supplementary materials.

Comment 22- Figure 3e needs to be annotated so the layers of the retina are made apparent to those unfamiliar. I am not convinced the hyporeflective lesions are very clear.

Response: Thank you. The annotation of layers added in Fig.3e and Figure legend. The hyporeflective lesions was remarked in Fig 3e as follows.

Comment 23- Line 197, the lesions are described as “progressive and highly significant” but there is no quantification of this.

Response: Thank you. The lesions are morphological change, which is difficult to make the statistical analysis. We changed the description from “progressive and highly significant” to “*remarkable*” (page 10 line 185).

Comment 24- Where are the data for the LC-MS/MS analysis?

Response: Thank you. This data was added in the supplementary Table3.

Comment 25- Same issue as before regarding the western blot in 5f.

Response: Thank you for your suggestion. I supplied the original fully annotated version in the supplementary materials.

Comment 26- Line 270, I don’t think there is evidence of normal transcription and translation. There is evidence of new mRNA transcripts and the western gives a hint of a larger protein. We do not know if this is the desired protein or how it compares to WT (at this point in the manuscript).

Response: Thank you for your comment. Regarding the mRNA expression, Sanger sequence verified it is the desired sequence, this data was uploaded in the supplementary materials and added on page 13 line 257-258 as follows: “*Sanger sequence confirmed the transcript sequence (Supplementary materials).*” Regarding the protein, new experiment was added. We constructed a plasmid containing the desired edited sequence i.e. pAV-CAG-CYP4V3(E1-6)-CYP4V2(E7-11)-bGH and transfected into NIH3T3 cells. Western blot analysis showed that this protein is in line with the larger protein in the treated retina, indicating normal translation in treated group. The related information was added in the Methods on page 24 line 468-470 as follow: “*Construct the pAV-CAG-CYP4V3(E1-6)-CYP4V2(E7-11) plasmid containing desired transcription sequence based on pAV-CAG-CYP4V2 from previous study²⁰. The fragment of CYP4V2(E1-6) was replaced by CYP4V3(E1-6) (NM_133969.3)*”, on page 27 line 521-525 as follows: “*NIH3T3 cells was transfected with 2.0 μg plasmid DNA of pAV-CAG-CYP4V3(E1-6)-CYP4V2(E7-11), together with 1 μg/μl polyethylenimine (PEI) transfection reagent (B600070, ProteinTech Group) according to the manufacturer’s protocol, and the ratio of DNA(μg) to PEI(μg) is 1:3. The control group are NIH3T3 cells without transfection. The medium was refreshed 6h post transfection and cells are harvested at 48h.*” and in the Results on page 13-14 line 257-260 as follows: “*Western blot demonstrated that two blots were shown in treated group, the shorter one is in line with the truncated protein from humanized mouse, and the larger one is in line with the blot from NIH3T3 cells transfected with plasmid pAV-CAG-CYP4V3(E1-6)-CYP4V2(E7-11) containing the desired edited sequence (Fig. 5f).*”

Comment 27- Line 281, has it recovered? That seems surprising as postoperative loss of ERG tends to be maintained over time. Is it possible that the untreated cohort has reduced to the level of functional reduction caused by an injection? Looking at the traces, I think if these were overlaid the empty vector injected cohort may be relatively stable from 2 months but the untreated lose function. How would it look if you plotted peak amplitudes at 1cdsm2 at each time point for each cohort?

Response: Thank you for your suggestion. we really appreciate it. Subretinal injection is an invasive operation, two vectors (2ul) were injected in this study and the bleb was absorbed one month later. Operation process has big influence on visual function evaluation. From this experiment we know the influence caused by operation is no longer statistically significant after 5 months. According to your suggestion, we revised the manuscript on page 14 line 275-277 as follows: “*At 5 months after treatment, the above differences were no longer statistically significant, indicating that the influence caused by operation can be regarded as negligible after 5 months.*”

There is no significant difference of plotted peak amplitudes at 1.0 cd*s/m2 at each time point for empty vector cohort with untreated cohort, the comparison of amplitudes at 1.0 cd*s/m2 for

high/low dose group with untreated cohort are added in the Supplementary Fig 2b.

Comment 28- Line 297, why would the high dose group show a significant reduction in ERG at 5 months but this become a significant improvement from 8 months?

Response: Thank you for your question. We discussed this in Discussion on page 20 line 381-390 as follows: "*Both Surgical operation and dosage affect the effectiveness evaluation⁵²...In this study, the visual function of the high-dose group was worse than that of the low-dose group in 5 months after treatment, while the opposite was observed at 11 months after treatment. The possible reason is that high-dose vector administration leads to overload of rAAV capsid protein and carrier DNA, thereby exceeding the processing capacity of retinal cells in the short term⁵¹, and after 8 months of treatment the stress may be counteracted through metabolism.*" Of course, if the dose exceeds the threshold irreversible damage will occur, more toxicology study needs to be taken in the future.

Comment 29- Line 304, I'm sorry, but I struggle to see any features here that are clearly defined and different between untreated and treated.

Response: I am sorry for any confusion. We added the annotation and asterisks to mark the difference in Fig 6b.

Comment 30- Line 351 onwards, I'm not sure it's relevant to mention the HMEJ work as this was not reported in the results.

Response: Thank you for your suggestion and I apologize for my redundant expression. The HMEJ work is the pre-experiment which has no correlation to this work, so we delete the HMEJ work in the discussion part (page 17 line 328).

Reviewer #2 (Remarks to the Author):

This is a very exciting research study by Meng et al. on the utility of HITI method of CRISPR/Cas9 in vivo gene editing. They have applied this technique to correct the most common genetic defect in the CYP4V2 gene-the causative agent for the degenerative eye disease Bietti's Crystalline Dystrophy (BCD). The efficacy of this system was demonstrated first using HEK cells followed by patient-derived iPSC cells differentiated to the appropriate ocular cell type-retinal pigmented epithelial cells. In vivo efficacy was finally demonstrated using a mouse model of BCD. There is great enthusiasm for this study and I only have minor comments which I will list below.

Comment 1. The authors state that "subretinal injection adversely affected on visual function..." What is the nature of this effect? Does retinal detachment occur? This needs to be clarified.

Response: We sincerely appreciate your valuable comments. Subretinal injection is an invasive operation, two vectors (2ul) were injected in this study and a big retinal detachment occur (more than 60% retina was detached), which has big influence on visual function. The bleb was absorbed one month later, visual function caused by retinal detachment is no longer statistically significant

after 5 months. This was revised in Discussion on page 20 line 382-385 as follows: “*Our results showed that subretinal injection caused retinal detachment, which has big influence on visual function evaluation, it is consistent with our clinical results of BCD gene replacement therapy²¹. The fluid was gradually absorbed, the influence caused by operation is no longer statistically significant after 5 months.*”

Comment 2. For the mouse study, 40 eyes per group were used. The Methods are unclear-did the authors use 20 mice per group or did they only inject one eye per mouse. The latter would be ideal in terms of safety data before initiating clinical trials. If the authors did not take this approach, justification should be provided.

Response: Thank you very much for pointing this out. Limited by the number of mouse model and considering the 3R principle, we used 20 mice per group (40 eyes) rather than 40 mice. This information was added on page 30 line 578-579 as follows: “*In accordance with the guidelines of 3R principle (Reduction, Replacement, Refinement) for laboratory animals, mice with bilateral eyes were included in the experimental group.*”

Comment 3. The authors state that only animals with no surgical complications were included for further study-what were the number of animals and did it differ between groups?

Response: We apologize for the confusion. Surgical process did have many complications including hemorrhage, which result in inaccurate vectors delivery. In order to exclude errors caused by surgical manipulations, those with successful operation were included for further study. N=40 is the number for further evaluation, there is no difference between different groups.

Comment 4. The quality/specificity of the CYP4V2 antibody used in this study is concerning. The blot in figure 1G has quite a bit of non-specific binding. On the other hand, the blot in figure 3C is quite clean and a different banding pattern is seen in figure 5F. To my knowledge, none of the mutations in the CYP4V2 gene produce detectable protein, including the H331P mutant <https://pubmed.ncbi.nlm.nih.gov/22772592/>. The authors need to address these discrepancies.

Response: Thank you for your comments. In Fig 1g, there are two type blots, one blot indicates CYP4V2 (60KD), while the other larger blot indicates CYP4V2-GFP fusion protein (85KD) produced by HITI editing. Both blots are specific. The same condition occurs in Fig 5f, the little one indicates truncated CYP4V3 (54KD) in *h-Cyp4v3^{mut/mut}* mice, while the other larger blot indicates the repaired CYP4V3 (60KD).

There are several articles demonstrated positive detection of CYP4V2 mutated protein. For example, the detection of mutated CYP4V2 protein carrying c.802-8_810del17insGC variant (Masayuki Hata et. al, *Proc Natl Acad Sci U S A.* 2018)

Fig. S4. (B) Western blot analyses of CYP4V2 protein levels in NOR and BCD iPSC-RPE cells infected with a mock-sequence (control), CYP4V2 WT, or CYP4V2 mut1 (c.802-8_810del17insGC).

Comment 5. The results of the in vivo studies are very encouraging, including reconstitution of ocular responses and lipid metabolites. Thus, one assumes that the RPE have been transduced, edited and are producing functional CYP4V2 protein. But, to be definitive, the authors should provide immunohistochemical staining for CYP4V2 in ocular tissue sections.

Response: Thank you for your valuable comment. In fact, mouse CYP4V3 protein shares 92% similarity with human CYP4V2 protein, and there is no specific antibody to mouse CYP4V3 and human CYP4V2. We failed to distinguish repaired CYP4V3 from endogenous mutated CYP4V3 by human-specific antibody in IF study. In addition, both mutated and repaired CYP4V3 protein has no different expression profile, we can't make the difference from IF and IHC. This is shown in the following figure.

Fortunately, the mutated and repaired CYP4V3 protein has different molecular weight, which helps to make a distinction by Western blot in Fig.5f.

REVIEWER COMMENTS

Reviewer #1 (Remarks to the Author):

Thank you for the consideration of my comments and improvements made to the manuscript, I appreciate the effort that has been made. The majority of my comments have been addressed and whilst perhaps not to my full satisfaction, I think the overall scope and encouraging results from this study make it worthy of publication. However, I do feel a couple of points raised that were not addressed should be included:

Where is the mCherry? The donor vector contained an mCherry reporter, retinal sections and flat mounts post-injection should be shown as this will help appreciate the scope of AAV8 transduction.

Also, with the use of an mCherry reporter, presumably cells could be processed by FACS to perform on-target editing assessments of transduced cells to compare to whole eye assessments?

I'm not sure Fig 5b is a reflection of editing efficiency (as labelled on the Y axis). It appears to be the population of mRNA transcripts carrying the edited sequence.

I do think NGS should be performed as this will be more accurate than subcloning.

Reviewer #2 (Remarks to the Author):

Comment 1 response: Acceptable

Comment 2 response: Acceptable

Comment 3 response: The authors need to provide the numbers of animals required to achieve their results. BCD patients and their caretakers may be misled into the translatability of the gene editing to the clinic. If there is a very high attrition rate then it should be clearly documented.

Comment 4 response: Unacceptable. The band the authors refer to as "mutant" CYP4V2 is likely non-specific. Particularly, the data from the 2018 PNAS figure shows a Western blot of

normal (NOR) and BCD iPSC-derived RPE cells. In NOR RPEs, there is no detectable full-length CYP4V2 but a prominent band for the “mutated” version-which should not be present in these cells. In fact, if one looks at the data for this antibody here <https://www.proteinatlas.org/ENSG00000145476-CYP4V2/summary/antibody>, there is no band detected on Western blot and immunocytochemistry is “The subcellular location is not consistent with literature” and immunohistochemistry states “Low consistency between antibody staining and RNA expression data.” This antibody has been discontinued and it would appear that is due in part to poor specificity, which can only be validated by adsorbing the primary antibody with the peptide immunogen.

Comment 5 response: Unacceptable. The IHC figure provided by the authors in comment 5 is of poor quality. The least that could be done is co-staining with another RPE marker like CD140b or RPE 65 to confirm cellular localization.

What is a “bugle” (lines 131-132)?

Response to Reviewer Comments/Questions

Our response to each comment/query is shown in blue text, all changes in the manuscript (marked in *red and italic*) are also shown throughout the following response letter.

Reviewer #1: Thank you for the consideration of my comments and improvements made to the manuscript, I appreciate the effort that has been made. The majority of my comments have been addressed and whilst perhaps not to my full satisfaction, I think the overall scope and encouraging results from this study make it worthy of publication. However, I do feel a couple of points raised that were not addressed should be included:

Comment 1: Where is the mCherry? The donor vector contained an mCherry reporter, retinal sections and flat mounts post-injection should be shown as this will help appreciate the scope of AAV8 transduction.

Response: Thanks a lot for your suggestion. Your comments are always constructive. The following picture is the results of flat mounts and retinal sections one month post injection. We added the scope of AAV8 transduction on page 12 line 227-230 and in the manuscript as follows: *“One month post injection, flat mounts results showed that mCherry fluorescence covered over 90% of retina. IF analysis of retinal sections indicated that EFS promoter drives strong gene expression in RPE cells as well as the outer and inner segment (OS/IS) and outer nuclear layer (ONL) of photoreceptor cells.”*

Comment 2: Also, with the use of an mCherry reporter, presumably cells could be processed by FACS to perform on-target editing assessments of transduced cells to compare to whole eye assessments?

Response: We greatly appreciate your suggestion, which will help to more accurately evaluate the on-target assessments. According to the mCherry fluorescence coverage results shown above, nearly 90% of retina was transduced. Our previous study (Liu X et al., eLife 2023) also shown cutting efficiency on those negative fluorescence cells, which was 32.58% (GFP+, n=5) vs. 11.04% (GFP-, n=5) at 3×10^9 dose. Based on the results in this manuscript, there is much difference in editing efficiency over 3, 6, 9 and 12 months. We plan to compare the transduced cells processed by FACS at the equal time points, however it will cost 15 months to finish this experiment. Though not accurate on-target assessments, the result from whole eye was able to reflected the editing tendency. We also added your precious suggestion in the discussion on page

19 line 368-370 as follows: *“Though highly effective virus infection was achieved in retina, the on-target assessments would be more accurate using dual viral vectors transduced cells rather than pan-retinal tissue.”*

Comment 3: I'm not sure Fig 5b is a reflection of editing efficiency (as labelled on the Y axis). It appears to be the population of mRNA transcripts carrying the edited sequence.

Response: Thanks for your suggestion. We agree that the data in Fig.5b is the population of mRNA transcripts carrying the edited sequence, which reflected the level of editing to some extent. We therefore changed it to *“proportion of edited mRNA transcripts”*. The related information is changed on page 12 line 231-232 and Fig. 5d in the manuscript.

Comment 4: I do think NGS should be performed as this will be more accurate than subcloning.

Response: We gratefully appreciate for your valuable comment. According to your suggestion, we performed the PCR-based NGS analysis of uninserted sequences and deleted our previous subcloning result. The length of PCR product for those sequences carrying new insertion is around 2kb, which is not suitable for PCR-based NGS examination (no more than 250bp), so we keep this result of subcloning (3000+ clones). We also uploaded the NGS results to the supplementary materials. We added it on page 13-14 line 257-263 as follows: *“For those sequences without insertion, we performed a PCR-based next-generation sequence (NGS) analysis in high-dose group at 11 months after treatment (Supplementary Fig.2a, Supplementary Table 3). The results showed that 4 large deletions of infidelity insertion occurring in 3' region introduced abnormal transcripts (4/2876) with AUGUSTUS prediction, other transcripts with/without insertion are normal (Supplementary Fig.2b). The corresponding translations with BioEdit revealed that 4 abnormal transcripts will result in non-functional protein production. (Supplementary Fig.2c).”* The methods and primers were added on page 26 line 497-500 as follows: *“In order to analysis those sequences without successful insertion in vivo, DNA extracted from retinal-RPE complexes of posttreatment 11 months mouse were amplified flanking target site by nested PCR (Supplementary Table 1). PCR amplicons were analyzed by NGS using the Illumina NovaSeq platform (Illumina, San Diego, CA, USA).”*

Reviewer #2

Comment 1 response: Acceptable

Comment 2 response: Acceptable

Comment 3 response: The authors need to provide the numbers of animals required to achieve their results. BCD patients and their caretakers may be misled into the translatability of the gene editing to the clinic. If there is a very high attrition rate then it should be clearly documented.

Response: Thanks a lot for your suggestion. We apologize for the improper statement that cause any confusion. Actually, the mice excluded from the following study are those with unsuccessful subretinal injection. It mainly caused by the surgical process, such as the retinal hemorrhage and vectors delivery failure. In order to exclude errors caused by experimental manipulations, only those animals with successful operation were collected for further study. The number of vitreous hemorrhage mice during the surgery is two. According to your suggestion, we revised the manuscript on page 31 line 605-606 as follows: *“Two animals with vitreous hemorrhage caused by operation were excluded, sixty animals with successful injection were included for further evaluation.”*

Comment 4 response: Unacceptable. The band the authors refer to as “mutant” CYP4V2 is likely non-specific. Particularly, the data from the 2018 PNAS figure shows a Western blot of normal (NOR) and BCD iPSC-derived RPE cells. In NOR RPEs, there is no detectable full-length CYP4V2 but a prominent band for the “mutated” version-which should not be present in these cells. In fact, if one looks at the data for this antibody here <https://www.proteinatlas.org/ENSG00000145476-CYP4V2/summary/antibody>, there is no band detected on Western blot and immunocytochemistry is “The subcellular location is not consistent with literature” and immunohistochemistry states “Low consistency between antibody staining and RNA expression data.” This antibody has been discontinued and it would appear that is due in part to poor specificity, which can only be validated by adsorbing the primary antibody with the peptide immunogen.

Response: Thanks a lot for your suggestion. I highly valued your suggestion and sincerely apologize for my misunderstanding of your previous concerns. We added those following experiments and wish the additional data can explain it clearly.

We identified the unique peptides of mutant CYP4V2 (carrying the following variant E7: c.802-8_810del17insGC) by nano HPLC-MS/MS (Thermo OE480) from cells derived from BCD patient. The spectrum of the unique peptide of mutant CYP4V2 is remarked as >mu. We also identified the unique peptides of mutant CYP4V3 (carrying the following two variants E7: c.802-8_810del17insGC and E8: c.992A>C) by nanoHPLC-MS/MS from *h-Cyp4v3^{mut/mut}* mouse, which remarked as >M265_mu.

>mu

181	MAGLWLGVLVQKLLWGAASALSAGASLVLSLLQRVASVARKWQMRPIPTVARAYPLV WQMRPIPTVAR PLV AYPLV	[88]
182	GHALLMKPDGREFFQIIIEYTEYRHMPLKLVWGPVPMVALYNAENVILTSSKQIDK GHALLMKPDGR GHALLMKPDGR EFFQIIIEYTEYR	[128]
183	SSMYKFLPWLGLGLLSTGNKWSRRKMLTPTFHFTILEDFLDIMNEQANILVKKLEKH	[180]
184	INQEAFCFFYITLCAIDIICETAMGNIGAQSNDDEYVRAVYRMESEIFRRIRKMPWLW NIGAQSNDDEYVR MSEMIFR	[242]
184	LDLWYLMFKEGWEHKSLQLHTFTNSGHDTTAAAINWSLYLLGSNPEVQKVDHELDDV KVDHELDDV VDHELDDV	[300]
185	FGKSDRPATVEDLKKLRYLECVIKETLRLEFSPVLEFARSVSEDCVAGYRVLKGTAVII FGK YLECVIK LFPSVLEFAR GTEAVII FGK SVSEDCVAGYR SDRPATVEDLK SDRPATVEDLKK	[360]
186	PYALHRDPRYFNPPEEFQERFFPENAGRHYPAYVPSAGPRNCIGQKFAVMEEKTILS PYALHR YFPNPEEFQER FAVMEEK FFPENAGR TILS	[420]
187	CILRHFWIESNQKREELGLEGLILRPSNGIWIKLKRRNADER HEWIESNQK EELGLEGLILR CTLR EELGLEGLILRPSNGIWK PSNGIWK	[480]

■ Carbamidomethyl[Q] 21
■ Oxidation[M] 4
■ Deamidation[N] 1

>M265_mu

[1]	MLWLWLGSLGQKLLWGAASAVSLAGATILISIFPMLVSYARKWQQMRSPSVARAYPLV	[60]
[8]	GHALYMKFNNAEFQQLIYYTEEFRHLPIIKLWIGPVPLVALYKAENVEVILTSSKQIDK AENVEVILTSSK	[120]
[12]	SFLYKFLQPWLGGLTSTGSKWRTRRKMMLPTFHFTILENFDVMNEQANILVNKLEKH	[180]
[16]	VNQEAFNCFYITLCAIDIICETAMGKNIGAQSNNSEYVRTVYRMSEMIFRRIKMPWLW NIGAQSNNSEYVR MSEMIFR	[240]
[24]	LDLWYLMFKEGWHEKKSLLQILHTFTNSGPDTTAAAINWSLYLGSNPEVQKKVDHELDDV KVDHELDDV VDHELDDV	[300]
[30]	FGRSHRPVTLEDLKKLYLDCVIKETLRVFPVPLFARSLSEDCVGGYKVTGTEAIIII FGR YLDCVIK VFPVPLFAR FGR SLSEDCVGGYK SHRPVTLEDLK	[360]
[36]	PYALHRDPRYPDPPEEFRPERFFPENSQGRHPYAYVFPFSAGPRNCIGQKFVMEEKTLA YFPDPEEFRPER FAVMEEK FFPENSQGR TILA	[420]
[42]	CILRQFVWESNQREELGLAGDLILRPNNGIWIKLRRHEDDP [463] QFVWESNQREELGLAGDLILRPNNGIWIK CILR	

For the quality of antibody. As shown in your query, <https://www.proteinatlas.org/ENSG00000145476-CYP4V2/summary/antibody>, the antibody has been suspended from selling over two years. This is why there are two types of antibodies used in our manuscript. The suspended antibody only used in our early cell experiments. Since then, we can't buy it anymore, and

therefore we explored several antibodies including the rabbit anti-CYP4V2 antibody generated by AbMax Biotechnology Co., Ltd. Immunogen CYP4V2 (NP_997235.3, 1 a.a. ~ 525 a.a) is the full-length human protein. In our first revision, we had replaced the WB results of Fig. 1g, 5f using the new antibody from AbMax. To avoid discrepancy, the remaining cell experiments (Fig.2f) which used the suspended atlas antibody was repeated with the AbMax antibody. We replaced the old figure (the left picture below) with the new one (the right picture below), and changed the suspended atlas antibody to AbMax antibody. We also added the Immunogen information in the manuscript. Thank you again for the reminder.

We also carried out immunofluorescent staining using the antibody from AbMax in RPE cells derived from normal control and two BCD patients who carrying homozygous c.802-8_810del17insGC and E8: c.992A>C variants respectively. The patient's genetic diagnosis are as follows:

As shown in the picture of immunofluorescent staining, the subcellular location stained with anti-CYP4V2 antibody (AbMax) is consistent with literature, which also supported the proteome analysis results shown above.

In addition, we also carried out IHC staining in retinal sections. Due to the influence of melanin in RPE cells, the local area was enlarged to make it clear. The positive dark brown staining (indicated by arrows) represented the CYP4V3 protein. It can be distinguished in WT, EV (*h-Cyp4v3^{mut/mut}*+empty vector), 1E9 and 3E9 groups, while absent in the NC (negative control) group. The IHC figure is also consistent with the immunofluorescent staining in Comment 5.

Comment 5 response: Unacceptable. The IHC figure provided by the authors in comment 5 is of poor quality. The least that could be done is co-staining with another RPE marker like CD140b or RPE 65 to confirm cellular localization.

Response: Thanks a lot for your suggestion. We performed co-staining of CYP4V3 and RPE65 to confirm the cellular localization with IHC. Because *h-Cyp4v3^{mut/mut}* mouse after treatment contained the mCherry, which may confuse the result of RPE65 (remarked in red), therefore we mainly performed the IHC on WT and *h-Cyp4v3^{mut/mut}* as follows:

Comment 6: What is a “bugle” (lines 131-132)?

Response: We apologize for the misspelling and change it from “bugle” to “*bulge*” in the manuscript. The following inserted figure is the schematic drawing of DNA bulge and RNA bulge.

We reference to other articles and found that genomic sites could be cleaved by CRISPR/Cas9 systems when DNA sequences contain insertions ('DNA bulge') or deletions ('RNA bulge') compared to the RNA guide strand, and Cas9 nickases used for paired nicking can also tolerate bulges in one of the guide strands. When explored the off-targets, we predicted all potential off-targets with no more than 3 mismatches to positions 4-20 of sgRNA3 with/without no more than one bulge in the genome.

REVIEWER COMMENTS

Reviewer #1 (Remarks to the Author):

Thank you for your continued efforts to address my comments. I'm sorry to make the same point but I am concerned about the lack of convincing evidence regarding the AAV transduction profile. I do not feel the images provided in the rebuttal are good enough for confirmation of this. The expression profile does not look at all as I'd expect in the retina, it seems more like over-exposure/incorrect image settings as the red is so uniform from the ONL to the RPE (OS signal is particularly surprising). Perhaps new injections can be performed and new eyes taken for whole mount and sectioning? Full-view C-sections would be helpful to show the transduction profile prior to zooming in for higher power images at highly transduced areas. Whilst the phenotype data seem encouraging, I feel it is critical to understand the vector transduction profile that achieves this.

I have also been re-considering the animal model and how to interpret the post-editing changes in phenotype.

New questions:

1. Were the C57Bl6J mice used for mouse model creation screened for rd8 and other rd mutations?
2. In the generation of the mouse model, were they screened for spontaneous mutations/off-target mutations?
3. Cyp4v3 detection in the retinal tissue is provided in the rebuttal to Reviewer 2 but should be included in the manuscript. If the protein is only in the RPE, why was AAV8 used? This comes round again to my query regarding the transduction profile. Cyp4v3 staining post-treatment should be included on tissue sections.
4. I think it would be important to perform SaCas9 staining on retinal sections when looking at mCherry expression.

Reviewer #2 (Remarks to the Author):

I appreciate the extensive work the authors have done to address my comments in the revised manuscript. In particular, the increased quality of the Western blot and immunocytochemistry for Comment 4 and the IHC for Comment 5. I have no further critiques and/or suggestions.

Response to Reviewer Comments/Questions

Our response to each comment/query is shown in blue text, all changes in the manuscript (marked in *red and italic*) are also shown throughout the following response letter.

Reviewer #1

Comment 1: Thank you for your continued efforts to address my comments. I'm sorry to make the same point but I am concerned about the lack of convincing evidence regarding the AAV transduction profile. I do not feel the images provided in the rebuttal are good enough for confirmation of this. The expression profile does not look at all as I'd expect in the retina, it seems more like over-exposure/incorrect image settings as the red is so uniform from the ONL to the RPE (OS signal is particularly surprising). Perhaps new injections can be performed and new eyes taken for whole mount and sectioning? Full-view C-sections would be helpful to show the transduction profile prior to zooming in for higher power images at highly transduced areas. Whilst the phenotype data seem encouraging, I feel it is critical to understand the vector transduction profile that achieves this.

Response: Thanks a lot for your suggestion. Your comments are really constructive. New injected eyeballs were extracted from sacrificed mouse mice at one month after treatment following removal of the anterior segments and lens for whole mount and frozen sections. For whole mount (a), we separated the RPE-choroid complex and cutting into four-leaf before photographing. Frozen sections prepared from eyecups, it includes full-view C sections (b) to show the transduction profile and zooming in for higher power images (c) to show the transduced retinal layer. The mCherry expressed negative (-) in the flat of lacking RPE cells due to stretching during preparation (top left) and the mount without transduced (low right). We added the related experiments and mouse information on page 12 line 229-234 in the manuscript as follows:

“One month post injection, flat mounts and IF analysis of full-view C sections results showed that mCherry fluorescence expressed widely in the retina (Supplementary Fig. 2a-b). Higher power images of retinal sections indicated that EFS promoter drives strong gene expression in RPE cells as well as the outer and inner segment (OS/IS) and outer nuclear layer (ONL) of photoreceptor cells (Supplementary Fig. 2c).”

Comment 2: I have also been re-considering the animal model and how to interpret the post-editing changes in phenotype. New questions: Were the C57BL6J mice used for mouse model creation screened for rd8 and other rd mutations?

Response: Thanks for your reconsideration. Indeed, a reliable animal model is important for the experiment. The much difference between C57BL/6N mice and C57BL/6J mice is that C57BL/6N mice harbor *CRB1^{rd8}* mutation while C57BL/6J mice do the opposite. To avoid the confusion in phenotype, we referenced the article (*Methods Mol Biol*, 2019) and tested commonly encountered *rd* mutations (*Pde6b^{rd1}*, *Crb1^{rd8}*, *Pde6b^{rd10}*, and *Rpe65^{rd12}*) in the mouse model. The results showed that there is no *rd* mutations in the C57BL6J mice model. We added the related information on page 9 line 175-177 as follows:

*“Commonly encountered retinal degeneration mutations (*Pde6b^{rd1}*, *Crb1^{rd8}*, *Pde6b^{rd10}*, and *Rpe65^{rd12}*) were excluded in the mouse model using sanger sequencing (Supplementary Table 1). “*

Comment 3: In the generation of the mouse model, were they screened for spontaneous mutations/off-target mutations?

Response: Thanks a lot for your suggestion. In the generation of the humanized knock-in mouse (*h-Cyp4v3^{mu/mut}*), a pair of sgRNAs targeting exon 6-8 of the *Cyp4v3* gene were designed. Through Cas-OFFinder (www.rgenome.net/cas-offinder/), total forty (twenty of each one) potential off-targets with no more than 3 mismatches to positions 4-20 of sgRNA3 with/without no more than one bulge were selected in the genome. Sequencing analysis of these targets failed to find off-target

sites. We added the related information on page 10 line 177-179, and uploaded it in the supplementary materials.

“Forty potential off-target sites through Cas-OFFinder were verified no detectable off-target activity using sequencing analysis (Supplementary Table 3).”

Comment 4: Cyp4v3 detection in the retinal tissue is provided in the rebuttal to Reviewer 2 but should be included in the manuscript. If the protein is only in the RPE, why was AAV8 used? This comes round again to my query regarding the transduction profile. Cyp4v3 staining post-treatment should be included on tissue sections.

Response: Thanks a lot for your suggestion. We added the Cyp4v3 staining of wild type mice, *h-Cyp4v3^{mut/mut}* and post-treatment mice in the manuscript on page 14 line 276-277.

IF analysis of retinal sections showed Cyp4v3 staining in the RPE cells (Supplementary Fig.3d).

For the reason of using AAV8 rather than AAV2, we comprehensively evaluated three aspects. Firstly, it is commonly reported that Cyp4v2 mutations affect lipid metabolism in RPE cells, leading to RPE dysfunction and subsequent photoreceptor degeneration (*Am J Ophthalmol*, 2016; *Hum Mol Genet*, 2004), which is consistent with our Cyp4v3 detection results. However, it is also reported that there is strong positive staining in RPE cells and weak staining in internal/external nuclear layers in the photoreceptors (*Mol Pharmacol*, 2012). Secondly, our previous study showed that serotype of AAV8 has higher transfection and expression efficiency to RPE and photoreceptor cells than AAV2 (*Journal of Peking University Health sciences*, 2020). We referenced to other articles as follows and also found the similar tendency (*Sci Transl Med*. 2011)

Thirdly, it is reported that neutralizing antibodies to AAV2 is the most common antibodies in all regions (*The Journal of infectious diseases*, 2009). On the contrary, AAV8 was isolated from nonhuman primates, and antibodies to AAV8 is expected to have low serum prevalence among humans.

Comment 5: I think it would be important to perform SaCas9 staining on retinal sections when looking at mCherry expression.

Response: Thanks for your suggestion. It is really important to show the SaCas9 expression. Just like the mCherry expression, the SaCas9 protein was also driven by EFS promoter. The mCherry expression was verified in RPE cells as well as photoreceptor cells (outer and inner segment -OS/IS and outer nuclear layer -ONL). We failed to observe the SaCas9 staining on retinal sections using the popular anti-CRISPR-Cas9 antibody (ab203943, Abcam). We referenced to other articles and found it difficult to do this staining in the retina sections. In addition, they commonly verified the SaCas9 expression using quantitative PCR to confirm Cas9 mRNA and gRNA expression and Western blot to reveal the stabilization of Cas9 protein (*eLife*, 2023; *Hum Gene Ther*, 2019). Therefore, we performed those experiments and added the results in the manuscript on page 12 line 234-236 as follows:

“Quantitative PCR confirmed efficient Cas9 mRNA and gRNA expression in two doses groups (Supplementary Fig. 2d-e, Supplementary Table 1) at 5 months after treatment, Western blot further revealed the stabilization of Cas9 protein in mouse retinal-RPE complexes (Supplementary Fig. 2f).”

REVIEWERS' COMMENTS

Reviewer #1 (Remarks to the Author):

I would like to thank the authors for their patience with me and for making such a respectful effort to address all my comments. I do believe this paper is now stronger for the additions and am happy to offer my recommendation to publish.